# Combined Approach: FFQ, DII, Anthropometric, Biochemical and DNA Damage Parameters in Obese with BMI ≥ 35 kg m^−2^

**DOI:** 10.3390/nu15040899

**Published:** 2023-02-10

**Authors:** Mirta Milić, Ivan Ožvald, Katarina Matković, Hrvoje Radašević, Maja Nikolić, Dragan Božičević, Lidija Duh, Martina Matovinović, Martina Bituh

**Affiliations:** 1Mutagenesis Unit, Institute for Medical Research and Occupational Health (IMROH), 10001 Zagreb, Croatia; 2Special Hospital for Extended Treatment of Duga Resa, 47250 Duga Resa, Croatia; 3Neuropsychiatric Hospital dr. Ivan Barbot of Popovača, 44317 Popovača, Croatia; 4Andrija Štampar Teaching Institute of Public Health, 10000 Zagreb, Croatia; 5Department of Internal Medicine, University Hospital Centre Zagreb, 10000 Zagreb, Croatia; 6Laboratory for Food Chemistry and Biochemistry, Faculty of Food Technology and Biotechnology, University of Zagreb, 10000 Zagreb, Croatia

**Keywords:** alkaline comet assay, micronucleus *cytome* assay, DII, FFQ Norfolk food questionnaire, obesity

## Abstract

Although obesity with its comorbidities is linked with higher cancer risk, the data on genome stability in the obese/severely obese are scarce. This is the first study with three DNA damage assessment assays (Fpg-modified and alkaline comet assays and micronucleus *cytome* assay) performed on a severely obese population (n = 53) where the results were compared with daily intake of food groups, nutrient intake, dietary inflammatory index (DII), and anthropometric and biochemical parameters usually measured in obese individuals. Results demonstrated the association between DNA damage levels and a decrease in cell proliferation with anthropometric measurements and the severity of obese status, together with elevated levels of urates, inorganic phosphates, chlorides, and hs troponin I levels. DII was connected with oxidative DNA damage, while BMI and basal metabolic rate (BMR) were associated with a decrease in cell proliferation and DNA damage creation. Measured daily BMR and calculated daily energy intake from the food frequency questionnaire (FFQ) demonstrated no significant difference (1792.80 vs. 1869.86 kcal day^−1^ mean values). Groups with higher DNA damage than expected (tail intensity in comet assay >9% and >12.4%, micronucleus frequency >13), consumed daily, weekly, and monthly more often some type of food groups, but differences did not show a clear influence on the elevated DNA damage levels. Combination of all three DNA damage assays demonstrated that some type of damage can start earlier in the obese individual lifespan, such as nuclear buds and nucleoplasmic bridges, then comes decrease in cell proliferation and then elevated micronucleus frequencies, and that primary DNA damage is not maybe crucial in the overweight, but in severely obese. Biochemically changed parameters pointed out that obesity can have an impact on changes in blood cell counts and division and also on genomic instability. Assays were able to demonstrate groups of sensitive individuals that should be further monitored for genomic instability and cancer prevention, especially when obesity is already connected with comorbidities, 13 different cancers, and a higher mortality risk with 7–10 disease-free years loss. In the future, both DNA damage and biochemical parameters should be combined with anthropometric ones for further obese monitoring, better insight into biological changes in the severely obese, and a more individual approach in therapy and treatment. Patients should also get a proper education about the foodstuff with pro- and anti-inflammatory effect.

## 1. Introduction

The World Health Organization (WHO) announced that up until now, more than 1 billion people worldwide are obese—650 million adults, 340 million adolescents, and 39 million children—and that by 2025, approximately 167 million people—adults and children—will become less healthy because they are overweight or obese [1]. Even the obese category has now three subcategories, mild (class I) with body mass index (BMI) ≥30 kg m^−2^ to <35 kg m^−2^; and severe (class II–III) with BMI ≥ 35 kg m^−2^ [2].

According to EUROSTAT 2019, the proportion of overweight adults varies across the European Union Member States with an average of about 53%, but in Croatia, this number is much higher, with 65% of adults with a BMI ≥ 25 kg m^−2^, and among them 42% who are overweight, and 23% obese [3].

Obesity is also associated with an elevated risk of several major non-communicable diseases (NCD), including type 2 diabetes, coronary heart disease, and several cancers [4,5,6]. There are different criteria in the estimation of the metabolic healthy obese persons, but if the most strict is used, only 10 % of obese (with BMI ≥ 30 kgm^−2^) can be categorized as metabolic healthy obese, and longitudinal studies demonstrated that even healthy obese adults show a strong tendency to progress to an unhealthy obese state [7,8,9]. Mild obesity is associated with the loss of 1 in 10 and severe obesity with the loss of 1 in 4 potential disease-free years during middle and later adulthood [10].

Severe obesity, characterized by excessive body fat mass, weight, and BMI, and its comorbidities, such as diabetes and cardiovascular diseases, contribute to increased genomic instability/cancer risk. There is strong evidence linking obesity with at least 13 sites of cancer: the esophagus, stomach, colorectum, liver, gallbladder, pancreas, breast, uterus, ovary, kidney, meninges, thyroid, and white blood cells [6]. Paradoxically, higher body weight has also been associated with reduced cancer risk—an obesity paradox that may be related to selection bias from reliance on the body mass index (BMI), which does not measure fat mass and fat-free mass [11]. It is not clear whether DNA damage is the effect of the disease nor can it predict obesity severity, further genomic instability, and cancer risk, and whether is it a useful additional parameter in obese severity prediction, and this aspect should be explored further.

In our previous studies, we already established the values for healthy populations for the two most common assays used in DNA damage assessment: the alkaline comet assay (which measures primary DNA damage and its modifications with enzymes, such as Fpg or formamidopyrimidine-DNA glycosylase, which also measure oxidative DNA damage) and the cytochalasin B blocked micronucleus *cytome* assay (CBMN, which measures different types of permanent DNA damage, cytokinesis, apoptosis, and necrosis frequency) on human fresh blood samples [12,13,14,15,16,17,18,19,20,21,22,23]. Studies with the same methods also demonstrated no indication for a threshold of a certain BMI for increased DNA damage, with the limit of up to 30 kgm^−2^, and BMIs higher than that value strongly correlated with DNA damage and even influenced changes or activation/deactivation of some types of DNA damage repair processes [21,24,25]. Few studies using one of these DNA damage methods demonstrated higher DNA damage levels in severely obese patients when compared to the controls, and those values were also higher when compared to our established values for the healthy population [20,24,26]. But still, the data are scarce for drawing final conclusions about the role of DNA damage in obesity, with most of the studies using only one DNA damage assay.

Since there are no studies conducted on severely obese people using the CBMN, alkaline comet, and Fpg-modified alkaline comet assay in the same study, we decided to check whether the use of those three assays together with biochemical and anthropometrical parameters, usually measured in obese patients, can be beneficiary for getting better insights in the obesity severity development, whether those values are higher when compared with the ones determined for the healthy population, and whether they can be useful additional biomarkers in the assessment of obese severity and in the detection of individuals more prone to develop genomic instability/tumor/cancer. Since both assays (CBMN and comet) can reveal patients more prone to cancer development, we focused only on metabolic unhealthy severely obese individuals, for whom we believed that by definition are more prone to develop comorbidities, if not already present when entering the study. We also used the European Prospective Investigation into Cancer (EPIC) Norfolk Food Frequency Questionnaire (FFQ) and their FETA software to study and analyze the influence of the diet and the dietary inflammatory index (DII) on the observed parameters [27,28,29,30,31,32]. It was found that the DII of the obese is associated with an increased risk of all-cause mortality, and among the obese with a BMI over 30, the DII showed a stronger connection with cardiovascular mortality [27,28,29,30,31,32].

We expect that the data collected in this research would be helpful in establishing more effective monitoring by providing insight into effective biomarkers for assessment of health risks in obese subjects, but also useful for future meta-analyses of the data on potential new biomarkers in NCD and DNA damage assessment. We believe that inclusion of the DNA damage assays in the severely obese biomonitoring can help in the detection of individuals more prone also to have a tumor/cancer, as a part of the comorbidities connected with obesity and NCDs.

## 2. Materials and Methods

If not stated otherwise, all chemicals and materials were from Merck (former Sigma-Aldrich, St. Louis, MI, USA).

### 2.1. Ethical Approval

The study gained ethical approval from Duga Resa Special Hospital for Extended Treatment, approval no.: 08-08-970/19); Institute for Medical Research and Occupational Health (IMROH), approval no.: 100-21/19-10; School of Medicine, University of Zagreb, Croatia, approval no. 380-59-10106-20-111/174, class: 641-01/20-02/01). Moreover, the study holds two Clinical Trials.gov numbers: NCT05055154 and NCT05007171.

### 2.2. Subject Recruitment, Food and Lifestyle Questionnaires

Volunteers (53 in total) were recruited in two hospitals (Special Hospital for Extended Treatment of Duga Resa and University Hospital Centre Zagreb) and at the IMROH.

The inclusion criteria were: BMI ≥ 35 kg m^−2^, the willingness to fill out all the questionnaires, and giving consent for anthropometric measurements and blood sampling. The study excluded subjects with any tumor/malignant disease at the time of sampling, those who passed any diagnostic procedure that included ionizing radiation within the last three months before the sampling; pregnant women, minors, and legally incapacitated persons. Considering that enrollment in the study was time-consuming (filling out the consent and the questionnaires on the spot) and that most of the participants were workers and were also obliged to come for sampling before their work, those were also exclusion/inclusion criteria, because we have taken only the volunteers in the study for whom we had all the necessary data and analysis results. Considering that besides BMI ≥30, all other criteria for the determination of metabolically healthy obese individuals would be determined after biochemical measurements, we recruited all the volunteers according to the previous criteria and decided to determine later, after the analysis, whether we have in our group metabolically healthy severely obese individuals [7].

Data on food consumption were collected by the standardized EPIC Norfolk FFQ. The same FFQ was used in EPIC studies for the assessment of the connection between cancer development and food intake (see Appendix A for FFQ details). The European Prospective Investigation into Cancer (EPIC) is an international collaboration studying diet and disease, with half a million participants [29]. The FFQ answers from volunteers about 130 commonly and less commonly used foods in the last year, divided into 9 categories (monthly: never or less than once; 1–3 times per month; weekly: once, 2–4; 5–6; daily: once; 2–3; 4–5; 6+), were coded according to the instructions of the software FETA designers and analyzed with the same software [28]. Results were analyzed in two ways to get the average daily food group intakes for 14 food groups (alcoholic beverages, cereals and cereal products, eggs and egg dishes, fats and oils, fish and fish products, fruit, meat and meat products, milk and milk products, non-alcoholic beverages, nuts and seeds, potatoes, soups and sauces, sugars-preserves and snacks, vegetables) on the group, sex and age level; and for the average daily intake of different ingredients calculated with FETA (alpha carotene (mcg), alcohol (g), beta carotene (mcg), calcium (mg), carotene—total (carotene equivalents) (mcg), carbohydrate: total (g), sugars (total) (g), fructose (g), galactose (g), glucose (g), glucose (mcg), lactose (g), maltose (g), starch (g), sucrose (g); cholesterol (mg), chloride (mg), copper (mg), Englyst fibre—non-starch polysaccharides (NSP) (g), iron (mg), total folate (mcg), carbohydrate, potassium (mg), energy: in kcal and in kJ, magnesium (mg), manganese (mg), sodium (mg), niacin (mg), phosphorus (mg), protein (g), vitamin A—retinol (mcg) and retinol equivalents (mcg), selenium (mcg), nitrogen (g), vitamins: B1—thiamin (mg), B2—riboflavin (mg), B12—cobalamin (mcg), B6—pyridoxine (mg), C—ascorbic acid (mg), D—ergocalciferol (mcg), E—alpha tocopherol equivalents (mg); zinc (mg); fat—total (g), monounsaturated fatty acids (MUFA—total) (g), polyunsaturated fatty acids (PUFA—total) (g), and saturated fatty acids (SFA—total) (g). FFQ was also served for diet inflammatory index (DII) calculation for 45 food ingredients with either pro- or anti-inflammatory potential.

Another questionnaire was used to collect personal data about lifestyle habits, diseases, therapies, work and leisure time, physical activity, smoking, exposures, education, workplace, hobbies, etc.—all the details that could also influence the results, especially on DNA damage. Answers to the questionnaire are also available in the Appendix A.

### 2.3. DII Calculation and Analysis

DII was calculated from the food questionnaire using formulas developed in 2009, and adjusted in 2014 and 2019. These formulas are calculated using 11 data sets from around the world [Australia (National Nutrition Survey), Bahrain (National Nutrition Survey for Adult Bahrainis), Denmark (Danish National Survey of Diet and Physical Activity), India (Indian Health Study), Japan (National Nutrition Survey Report), Mexico (Mexican National Health and Nutrition Survey), New Zealand (National Nutrition Survey), South Korea (Korean NHANES), Taiwan (Nutrition and Health Survey in Taiwan), the United Kingdom (National Diet and Nutrition Survey), and the United States (NHANES)] that formed the basis for calculated DII mean and standard deviation (SD) values for 45 food parameters [30,31]. Among them, 9 parameters were pro-inflammatory (energy, proteins, total fats, carbohydrates, saturated fatty acids, trans fatty acids, cholesterol, iron, and vitamin B_12_) and the other 36 had anti-inflammatory properties (monounsaturated fatty acids, polyunsaturated fatty acids, ω-3 fatty acids, ω-6 fatty acids, dietary fiber, alcohol, vitamins A, D, E, C, and B6, β-carotene, thiamin, riboflavin, niacin, folate, Mg, Se, Zn, flavan-3-ols, flavones, flavonols, flavonones, anthocyanidins, isoflavones, caffeine, garlic, onion, pepper, oregano, rosemary, eugenol, saffron, ginger, and turmeric).

In detail, nutritional components for the calculation of inflammatory dietary indices were taken from databases containing the chemical composition of food and beverages. Depending on the method of preparation, the values for flavan-3-ols, flavones, flavonols, flavonones, anthocyanidins, and isoflavones in raw foods were multiplied by retention factors. The retention factor for cooking is 0.59, for frying 0.5, and for baking 1.09 [33]. For each food component, the specific inflammatory index of the individual component was first calculated in such a way that the global average intake for the calculated food component was subtracted from the obtained average value of the intake of the food component, and the thus obtained value was divided by the standard deviation of the calculated individual food component, which resulted in a specific z-value. The obtained z-value of the food component was converted into percentiles centered at zero and doubled. A value of one was subtracted from the thus obtained value, which was ultimately multiplied by the value of the pro-inflammatory or anti-inflammatory effect of the calculated component. The inflammatory index of the diet was obtained by adding up all 45 inflammatory indices of individual components. The ranges of the obtained values must be between 7.98, which represents the maximum pro-inflammatory effect, and −8.87, which represents the maximum anti-inflammatory effect of the diet. Values for the global average intake, standard deviations, and pro-inflammatory/anti-inflammatory effects of all 45 food components as well as the method of calculating the DII were taken from the papers of Shivappa et al. (2014) and Hebert et al. (2019) [30,31].

### 2.4. Anthropometric Analysis

All participants were measured for their height with the stadiometer. Another anthropometric analysis was performed with a body composition analyzer (InBody 270, InBody, Seoul, Korea) according to the manufacturer’s instructions. Analysis was performed by the medical personnel in the hospital/Institute for Public Health mentioned for volunteer recruitment on the same day as the blood sampling.

The analyzer gave information about body composition (total amount of water in body, amount of protein in kg for muscle building, amount of minerals in kg for strengthening bones, amount of body fat mass in kg for storing excess energy, weight in kg as the sum of all other parameters in this group), muscle-fat analysis (weight in kg, skeletal muscle mass-SMM in kg, body fat mass in kg—BFM), obesity analysis (body mass index—BMI and percentage of body fat -PBF in %), segmental lean analysis, segmental fat analysis, weight control (kg), waist-hip ratio (W-H ratio), visceral fat level (VFL), and research parameters (fat-free mass in kg, basal metabolic rate—BMR kcal day^−1^, obesity degree, recommended calorie intake per day), and InBody score.

### 2.5. Sampling

Volunteers gave 15 mL of whole blood in different tubes: for serum biochemical parameters (tubes with a silica clot activator) and for DNA damage assessment (heparin tubes and EDTA tubes) (Becton Dickinson, Franklin Lakes, NJ, USA).

### 2.6. Biochemical Parameters Analysis

The clinical chemistry autoanalyzer Beckman Coulter AU 480 (Beckman Coulter, Inc., Brea, CA, USA) was used to determine the levels of glucose, urea, urates, inorganic phosphates, chlorides, albumins, C-reactive proteins (hsCRP), total cholesterol (TC), HDL-C, LDL-C, and triglycerides (TG). The immunoassay analyzer ARCHITECT i1000sr (Abbott, Chicago, IL, USA) determined the levels of insulin, hs troponin I, TSH, fT3, and fT4 levels. Insulin resistance as HOMA-IR was calculated according to the formula: glucose (mmol L^−1^) × insulin (mIU L^−1^)/22.5. An automated hematology analyzer (XS-1000i) with an autosampler (Sysmex, Kobe, Japan) was used to determine the complete blood count (CBC).

### 2.7. Alkaline and Fpg Comet Assay

For the alkaline and Fpg comet assays, 2 mL of whole fresh human blood from each volunteer was collected in an EDTA vacutainer, transported to the Institute for Medical Research and Occupational Health at 4 °C, light protected, on the same day, and immediately processed further. The procedure for the alkaline comet assay was described previously [34], and details can also be seen in the compendium of protocols (Nature protocols) [35], but we will repeat it here again. Comet slides were already prelayered with 300 μL of 1% normal melting point (NMP) agarose and dried. Ten microliters of blood aliquots was mixed with 100 μL of 0.6% low melting point (LMP) agarose. Finally, 70 µL of cell/gel mixture was used and a 2 gel system with 24 × 24 mm glass cover slides (Vitrognost, Biognost, Zagreb, Croatia) was included, and all the samples were performed in duplicate. Mixtures put on comet slides were covered with a coverslip. After 10 min at +4 °C, coverslips were removed and slides were transferred into the freshly prepared cold lysis solution (2.5 M NaCl, 100 mM Na_2_EDTA, 10 mM Trizma base, 1% Triton, 10% DMSO, 1% Na lauryl sarcosine, pH 10) at +4 °C and kept there for 1 h at +4 °C. After lysis, slides were gently washed with distilled water and kept for 20 min in the freshly prepared cold denaturation solution at +4 °C (300 mM NaOH, 1 mM Na_2_EDTA, pH > 13), and then placed into a horizontal electrophoresis tank in a new denaturation solution at +4 °C for a further 20 min during electrophoresis at 0.8 V/cm and 300 mA. All of the steps were performed under dim light. After three washes in a neutralization step with 0.4 M Tris-HCl (pH 7.5), for 5 min each, slides were immediately processed further, stained with ethidium bromide at a concentration of 20 μg/mL, and analyzed under the Olympus BX51 (200× magnification) fluorescence microscope using the Comet Assay IV software for image analysis (former Perceptive Instruments, now Instem, London, UK). A total of 100 nucleoids (50 per slide) were measured per sample. The tail intensity parameter (TI, % of DNA in comet tail) was used. Samples from both gels of the same individuals were compared, and details will be explained in the statistical part.

The blood samples in the same agarose concentration (70 µL of cell/gel mixture) and 2 gel system were also used for the Fpg comet assay. In this assay, we had more slides, since duplicates should be performed for the alkaline comet assay with the treatment with only buffer F and duplicates should also be performed for the Fpg enzyme plus buffer F treatment. Up to the lysis, all the steps were the same as for the regular alkaline comet assay. After lysis, slides were immersed in 3 changes of Buffer F (40 mM HEPES, 0.1 M KCl, 0.5 mM Na_2_EDTA, and 0.2 mg L^−1^ BSA, pH 8.0, Sigma-Aldrich, USA) for 5 min each time. Fpg gels were treated with 70 µL Fpg enzyme each (Norgenotech, Oslo, Norway), diluted in Buffer F, while controls were treated only with 70 µL Buffer F and covered with coverslips. After 60 minutes of humid chamber slide incubation at 37 °C, coverslips were removed, gels were washed with distilled water, and then followed the procedure as described for alkaline comet assay, but all the equipment, treatments, lysis, denaturations, electrophoresis, and neutralizations, were performed separately from the regular alkaline comet assay to avoid any influence on the results.

Slides were scored under an epifluorescence microscope (Olympus BX51, Tokyo, Japan), under 200× magnification. The level of DNA damage in individual cells was assessed with Comet Assay IV^TM^ software (Instem-Perceptive Instruments Ltd., London, UK). The descriptor of interest was tail intensity (TI, % of the genomic DNA that migrated during the electrophoresis from the nuclear core to the tail), measured in agarose-embedded cells (damaged ones had the shape of a small/longer-tailed comet). Fpg TI Net value was calculated after subtraction from the Fpg control buffer values [36]. All samples were conducted in duplicate, and in total, 100 comets were analyzed for each individual and assay (50 in each duplicate that were also compared before further analysis). According to [36], the means of two medians were used for the final analysis and results.

For both the alkaline and Fpg-modified alkaline comet assay, we did not use internal control in this study. In our previous studies, collecting databases on more than 19,000 individuals all over the world, we have determined the reference values for TI values for the healthy population of up to 9%, the cancer-free population of up to 12.4%, cancer-prone/cancer cases 17.7%, and deceased population 18% [17,18]. The age of the entire group was a working population, and special analyses were performed on the children, teenagers, and retired workers groups; they were not included in the main analysis. Body mass index was not reported for all studies, but we previously mentioned that it was shown that a significant influence on the level of DNA damage measured with TI parameter starts when BMI is higher than 30 kg m^−2^ [24,25].

### 2.8. Micronucleus Cytome Assay

Details of the method can be seen in our previous publication [37], but we will describe it here also. For the CBMN assay, blood samples collected by venipuncture into heparinized tubes (Beckton Dickinson, Franklin Lakes, NJ, USA) were coded, stored at 4 °C, and transferred within 2 h to IMROH, where they were immediately embedded in a cell culture medium. For each sample, duplicates of the cell culture were prepared in a 25 cm^2^ flask through the addition of 0.6 mL of heparinized blood under sterile conditions into a prewarmed cell culture medium. The cell culture medium consisted of 6 mL RPMI-1640 medium, 1 mL fetal calf serum (FCS), 0.02 mL phytohemagglutinin-L, and 0.01 mL antibiotic solution (100 IUmL^−1^ penicillin and 100 mg mL^−1^ streptomycin) and was prepared freshly and kept at 37 °C for at least 1 h before cultivation. Cultures were successively incubated at 37 °C with 5% CO_2_ in the air in a humidified atmosphere in the cell incubator (Thermo Fischer Scientific Inc., former Heraeus, Langenselbold, Germany). After 44 h of incubation, cytochalasin-B was added to the cultures in sterile conditions at a final concentration of 6 mg mL^−1^ for cytokinesis blocking. After 72 h of incubation, cells with the medium were transferred into glass centrifugation tubes, which were run at 450× *g* for 10 min in a centrifuge (Rotofix 32a, Hettich, Tuttlingen, Germany) using a swing bucket rotor. Then, the solution was removed, and the cell pellet was gently mixed with a cold, mild hypotonic solution (75 mM KCl) and left at room temperature for 10 min. After supernatant removal, the cell pellet was fixed with a fresh mixture of cold methanol/acetic acid (3:1 *v/v*) (Kemika, Zagreb, Croatia). The treatment with the fixative was repeated three times, and the cell pellet, dissolved in a minimal volume of fixative, was seeded on clean, cold microscopic slides (Vitrognost, Zagreb, Croatia), dried, and stained for 10 min with 5% Giemsa (pH 6.8) prepared freshly in distilled water (Yasenka, Vukovar, Croatia). Microscope analysis was performed at 400× magnification using a light microscope (Olympus, Tokyo, Japan). MNi (a biomarker of chromosome breakage and/or whole chromosome loss), nucleoplasmic bridges (NPB, a biomarker of DNA misrepair and/or telomere end-fusions and bicentric chromosomes), and nuclear buds (NB, a biomarker of elimination of amplified DNA and/or DNA repair complexes) were scored in 2000 binucleated lymphocytes with well-preserved cytoplasm per subject. A total of 1000 lymphocytes per donor were scored to evaluate the frequency of cells with 1–4 nuclei (M1, M2, M3, and M4) within the same cytoplasm. The cytokinesis-block proliferation index or nuclear division index (NDI) was calculated according to the following formula, with M1–M4 representing the number of cells with 1–4 nuclei, and N the total number of cells scored (1000) [38].
NDI = (M1 + 2M2 + 3M3 + 4M4)/N

The frequency of apoptotic (programmed cell death) and necrotic cells in 1000 lymphocytes per subject was also scored, and the scored DNA damage parameter frequency (per 2000 binucleated lymphocytes (BN)) was calculated and also expressed as 1000 BN.

For this study, we did not use an internal control for the assay. In our previous study, we established the reference MN values for a healthy Croatian population based on 200 individuals (up to 12.5 or 13 MN per 1000 BN), which also corresponded to the HUMN group, and according to the HUMN group, reference values for NB (up to 5 per 1000 BN) and NPB (up to 10 per 1000 BN) [14,15,16,39]. The HUMN group established those values based on the results of almost 7000 individual databases collected worldwide, in which we also participated. In the reported study, the mean age of the group was 44, with almost equal participation of females and males, and the BMI index was not included in the published results, but a significant difference in values was seen in the group of 40 years and older [14,15,16,39].

### 2.9. Statistical Analysis

Using the Statistica^®^ 13.05.0.17 software package (TIBCO Software Inc., Palo Alto, CA, USA) for the analysis, descriptive statistics were carried out (mean ± standard deviation (SD), standard error (SE), and median, range). The EPIC-Norfolk food questionnaire was analyzed with FETA software [28]. Anthropometric, biochemical, DII, DII food parameters, FFQ parameters, DNA damage, and oxidative damage parameters were analyzed after descriptive statistics using the Mann–Whitney U test. The mean and median values of the values obtained by the standard and Fpg-modified comet assay, the mean of two medians, were analyzed with the Mann–Whitney U test on the group level. Univariate analyses were performed by Spearman’s nonparametric correlation test. In the comet assay, Fpg comet assay, and CBMN assay, duplicate slide/gel results were compared with ANOVA, and after not finding any differences between them, the final results were calculated after the duplicate grouping into one sample per individual. The statistical significance threshold was set at *p* < 0.05. Except for statistical significance, all data with a decimal point were set to two decimals. For DNA damage modeling for TI values in the comet and Fpg comet assays and for MN, NB, and NPB values in the MN cytome assay, we used the free software R statistical programming language (version 4.1.2, R Core team and The R Foundation, Indianapolis, IN, USA). The stepwise regression model was performed to investigate the relationship between different features that can affect measured tail intensity, MN, NB, and NPB and give us a model prediction for the used anthropometric and biochemical parameters. Stars used in these graphs are *—*p* value < 0.05, **—*p* value < 0.01, and ***—*p* value < 0.001.

## 3. Results

We wanted to investigate the relationship between different biomarkers on the group level, then on the gender level, and finally on the age level (with different age categories). Since those categories can possibly influence the results, and establish how dietary inflammatory index and food ingredients are connected to inflammation and influence our results and individual DNA damage results, the results are also grouped in that way. This manuscript is supposed to serve as the basis for future meta-analyses, but since the data are huge and it is confusing to navigate through all the data, we will present the main findings in the main manuscript and the whole tables will be in the Appendix A.

Individual demographic details can be seen in Appendix A.

In general, we had 36 females and 17 males. There were only 3 smokers; two of them smoked the same number of cigarettes per day (one for 30 and the other for 4 years), while the third person smoked electronic cigarettes. In the entire group, 18 subjects were highly educated (university degrees), and the rest had middle or lower school finished. Seven subjects declared occasional exposure to pesticides (2–3 times per year, during gardening, field, or orchard work), and the other two reported twice-yearly exposure to wood or metal varnish (due to hobbies and activities they do in their free time).

Among them, 25 individuals had closer family relatives with tumor/cancer diseases. Eight participants had a tumor surgically removed in the past. Hypertension was diagnosed among 28 individuals, 5 had asthma, 19 were with diabetes type II, 4 with hypothyroidism, and 18 with dyslipidemia. Some of them occasionally used vitamins or supplements, but not regularly. Eighteen of them were physically active (walking to their workplace, everyday walking or walking 3–4 times per week, or light hiking once per week).

Considering medication, 30 reported the use of antihypertensives, 17 the use of antileptics, and 17 the use of antidiabetics.

As for the age of the group, mean, median, SD, SE, and range were: 51.15 years; 51.41; 11.59; 1.59; 26.35–68 years.

The results reported in Table 1 show that the entire group had elevated levels of all anthropometric parameters when compared to referent values for a healthy and non-obese population. Although there was also an increase in skeletal muscle mass, elevated levels of weight and BMI were mostly related to increased fat levels and body fat mass, especially visceral ones. Body fat mass for the entire group was at least 6-fold higher than the normal reference range values.

Weight changes correlated with BMI, BMR, BFM, and SMM increases, but there was a very weak, although still significant, correlation between BMR and BMI.

Next, we wanted to investigate the trends in the values of biochemical parameters (Table 2). We should mention immediately that we realized after the performed analysis that none of the individuals included in our study was metabolic healthy severely obese individual, not only by the strict Karelis categories [40] (levels of TG, HDL-C, HOMA-IR, TC, and LDL-C) but also by the other less strict categories that were mentioned in the review by Tsatoulis and Patsou [7].

CBC was, according to the mean and median values, in the range of reference values, although for each parameter there was at least one person who had higher levels than the reference range. Lipid profile (TC, LDL-C, TG), glucose and HOMA-IR, hsCRP, and urates were higher than the reference range of values.

Looking at an individual level, 20 individuals had a lower mean corpuscular volume of red blood cells (MCV) values; 16 of them had higher neutrophil counts; 25 had higher glucose levels; 31 had higher urate values; 34 had higher hsCRP values; 34 had higher TC values; 33 had higher LDL-C levels; 32 had higher TG values; 11 had higher insulin levels; and 48 had higher HOMA-IR values than the reference values. In 3 individuals, TSH levels were above the reference range.

In Table 2, we demonstrated only the correlations that were moderately and strongly affected within the correlation analysis, and all correlations found can be seen in the Appendix A). Here, we will focus only on mentioning the results that were either higher or lower than the reference values on the level of the entire group or among the highest number of individuals within the group.

From the elevated levels of biochemical parameters, only a few of them correlated with the anthropometric ones. Insulin levels moderately correlated with weight; urate levels moderately correlated with SMM and BMR; and HOMA-IR levels moderately correlated with weight and BMI.

TC demonstrated a very strong correlation with LDL-C and moderate with TG, HDL-C, and hs troponin. Hs troponin is a biomarker of myocardial injury and independently associated with obesity and BMI [41]. Therefore, this was an interesting finding, although with regardless of only a moderate level of correlation. The same moderate correlation level of hs troponin was also found for LDL-C, urate levels, and glucose. HOMA-IR very strongly correlated with insulin, strongly with glucose, and moderately, but negatively, with MCV. Considering that almost half of the group had lower MCV, and with a moderate correlation with HOMA-IR, this was also an interesting finding, since lower MCV levels are associated with a higher risk of metabolic diseases [42]. MCV also demonstrated a moderate correlation with insulin levels. Besides obvious correlations between neutrophil counts and other cell types in the blood, their counts are also moderately correlated with hsCRP levels. Since obesity is usually accompanied by a mild, chronic, systemic inflammation, and considering that, neutrophils will be the first to infiltrate adipocytes in adipose tissue in fat accumulation [43], and almost half of the individuals had higher neutrophil levels, this is also an interesting finding.

Although levels of inorganic phosphates and chlorides were within the reference range, we will mention them here since in this table and also in the table for genomic instability, they showed a slightly moderately positive correlation with DNA damage levels, elevated anthropometric parameters, and also hs troponin. Although the correlation is moderate but present in all tables, this is also an interesting finding that should be further explored with a larger number of individuals to get some final answers. Inorganic phosphate has been associated with different chronic diseases, including type 2 diabetes mellitus, obesity, and even cancer, and linked with tissue damage and antiproliferative effects depending on the cell status [44].

Table 3 demonstrates DNA damage parameters measured by three different assays. With both Fpg-modified and alkaline comet assays, the measured group mean values were higher than the reference range. The maximum levels estimated for people with high cancer risk (alkaline comet assay) and a very high percentage of oxidative DNA damage (Fpg comet assay) strongly suggest that there should be further monitoring on the individual level. Again, we showed only moderate, strong, and very strong correlations, and the rest can be found in the Appendix A).

MN *cytome* assay demonstrated a decrease in nuclear proliferation index (less than 2), with more mononuclear (M1) than binuclear cells (M2). Among the subjects, higher frequencies for apoptotic cells were also noticed. Although the group mean value of micronucleus frequency was in the reference range, there were individuals with higher levels of damage. The same goes for the NB and NPB parameters, again pointing out that there should be further analysis at the individual level.

Briefly, the results demonstrated that there was a moderately negative correlation between M2, M3, and M4 cell frequency, and NDI with the elevated anthropometric parameters of weight, BMI, and BFM. A positive correlation was seen with M1 cell frequency. This means that the increase in weight, BMI, and BFM in the severely obese is influencing the lowering of the mitotic ratio of the cells counted in the micronucleus assay. Similar was seen for MN mitotic parameters and levels of inorganic phosphates, chlorides, and hs troponin, although the correlation was moderate. Inorganic phosphates, chlorides, and hs troponin, but also glucose levels, had a moderate correlation with the level of necrotic cells. Alkaline comet and micronucleus cytome assays did not show any correlation with other parameters. In the alkaline comet assay, TI had a moderately positive correlation with TG and chlorides, while weak correlations were observed for apoptosis, glucose, and inorganic phosphates.

From the FFQ questionnaire, a refined scoring algorithm for DII was used to classify 45 food items that scored in a range from 7.89 (strongly pro-inflammatory) to −8.87 (strongly anti-inflammatory). The “Food parameter-specific DII scores” are then summed to create the “overall DII score” for an individual and for the group. For the entire group, the DII mean, median, minimum, maximum, and standard deviation values were: 2.06, 2.18, −3.37, 6.73, 2.39, and 0.33. DII demonstrated a weak positive correlation with biochemical parameters: leukocytes (0.31), neutrophils (0.27), lymphocytes (0.27), monocytes (0.29), and a negative correlation with hsCRP values (−0.29). DII correlated with micronucleus *cytome* assay values (MN and apoptosis, Figure 1). DII did not correlate with the standard comet assay but demonstrated a weak positive correlation with the Fpg-modified comet assay (R = 0.28, Figure 2).

Daily consumption in grams for 130 food ingredients in the last year from the FFQ questionnaire on the group levels is presented in Table 4 (12 non-overlapping food categories) and Appendix A (since all the correlations were weak, we have moved the tables into Appendix A).

On a daily basis, in the entire group, there were 23 consumers of white bread and rolls (8 once, 12 twice or thrice, and 3 4–5 times daily), 13 of whole meal bread and rolls (5 once, 7 twice or thrice per day, and one 4–5 times daily), and 12 of brown bread and rolls (4 once and 8 twice or thrice daily). There were 34 daily coffee with caffeine consumers (9 once, 24 twice, or thrice, and one 4–5 times daily) and 16 tea consumers (10 once, 5 twice or thrice, and one 4–5 times daily) and among them, there were 12 who were daily adding sugar to coffee/tea. Ten eat chocolate regularly daily (4 once and 6 twice or thrice). Seven eat cheese (5 once and 2 twice or thrice daily). Of course, there were 13 regularly consuming garlic daily (9 once, 5 twice or thrice), 15 onions once per day, 13 consumed apples daily (9 once and 5 twice or thrice), 7 oranges (6 once per day and one twice or thrice) and 7 consumed once per day a banana.

On a weekly basis, there were 25 consuming beef (10 once, 14 twice or 4 times, and one 5–6 times weekly), 31 pork consumers (17 once, 11 2–4 times, 3 5–6 times), 43 chicken (34 once, six 2–4 times), 25 bacon (18 once, ten 2–4 times) and 24 ham weekly consumers (13 once, six 2–4 times, five 5–6 times weekly). Considering potato consumption, there were 38 boiled potato consumers per week (11 once, 22 2–4, five 5–6 times), 21 pommes frites (13 once, five 2–4 times, three 5–6 times per week), 33 roast potato consumers (28 once, three 2–4 times, two 5–6 times), and 22 potato salad consumers (19 once and three 2–4 times per week). There were 35 weekly egg consumers (11 once, 20 2–4 times, and 4 5–6 times per week), 30 consumed cottage cheese (15 once, 11 2–4 times, 4 5–6 times per week), and 26 cheese weekly (8 once, 11 2–4 times and 7 5–6 times per week). Among them, there were 29 weekly white rice consumers (21 once, 7 2–4 times, 1 5–6 times weekly) and 32 white pasta consumers (20 once, 11 2–4 times, and 1 5–6 times per week). Thirty-three consumed meat soup weekly (18 once, 13 2–4 times, 2 5–6 times per week), 16 ice cream (11 once, 5 2–4 times weekly), 18 chocolate (9,7,2), 10 chocolate bars (5,4,1), and 14 ready-made buns (9,4,1). There were also 25 consumers of low-fat yoghurt (9,10,6) and 14 full-fat yoghurt consumers (8,4,2).

Results reported in the tables demonstrated that older individuals lower their energy intake, sugar intake, and alcoholic beverage consumption. Proliferation parameters and less damage in the comet assay correlated with vegetable intake. Damage levels measured by the comet assay and the Net Fpg comet were negatively correlated with cereals and cereal products and preserves and snacks. Micronucleus frequency was also correlated with the consumption of eggs and egg dishes. The inflammatory dietary index was negatively correlated with the intake of cereals and cereal products, fish and fish products, fruit, and milk. TG negatively correlated with consumption of the healthy fats and oils. Vitamin C negatively correlated with the Net Fpg comet assay values and proliferation parameters recorded by the micronucleus assay.

The dietary inflammatory index for the entire group did not demonstrate high DII at the group level, which was a good fact since it is highly recommended that DII be as low as possible. However, when individual DII was calculated (Figure 3), there were individuals with very high DII values and even those with negative values. Negative values are beneficial since it is a well-known fact that a low-DII diet has an anti-inflammatory effect.

When divided into two groups by sex (Appendix A), the intake of food groups did not make a significant difference. It was noted (marked in black) that consumption of some nutrients was more prevalent in the female group (fibers, all types of carbohydrates, energy intake, magnesium, sodium, niacin, phosphorous, proteins, selenium, vitamins C and E, zinc, fat, MUFA, PUFA, and SFA).

When we analyze the differences in the consumption of certain food groups by sex, only the intake of nuts and seeds was significantly different (*p* = 0.04, more in the female group). However, males used more alcoholic beverages, and females used more cereals, milk products, healthy oils, fruits, vegetables, preserves, and snacks. 

When we talk about other biochemical and anthropometric parameters and significant differences, females had significantly less hs troponin I but more TC, HDL, HOMA-IR, higher weight, BFM, PBF%, and W-H ratio. Males had significantly higher values for SMM and BMR. Both groups had similar W-H ratios and BMIs, but males had higher BMRs.

Both groups did not significantly differ in DII values (but it was higher in males), micronucleus, or comet assay values, but the proliferation index was higher in the female group. The female group had generally lower damage measured with micronucleus and comet assays, except for the oxidative damage (Fpg), which was higher in that group.

After the subjects were divided into different age groups, we found some significant values (Appendix A). We decided to use similar age categories as in our previous study since those divisions demonstrated higher DNA damage in some groups (e.g., MN and NB frequencies in the >60 age group, NPB levels in the <60 age group, or NPB levels in the 41–50 age group) [37], and we wanted to check what is happening in this group of obese individuals, not only on DNA damage levels but in all other parameters.

When divided into groups with ages ≥60 and <60 (Appendix A), subjects ≥60 years of age had significantly higher levels of MN frequency, hs troponin, sugars (g), intake of alcoholic beverages, level of saturated fatty acids (SFA), carbohydrate sucrose, carbohydrate maltose, higher BMI, TSH, and DII, and they ate more vegetables.

The group of subjects younger than 60 was further divided into subgroups of 51–59 years, 41–50 years, and <40 years (Appendix A), and their results were intra-compared and also compared with subjects ≥60 years. We found that groups 51–59 and ≥60 years differed significantly (higher values for ≥60) in proteins, SMM, and BMR. All other food groups’ intake and biochemical and anthropometric parameters were higher in the “younger” group (marked in bold). It seems that subjects younger than 60 had a more diverse diet, bigger portions, and even a higher BMR than the elderly group. Further, the diet in the “elder” group had a higher DII index; these subjects had a higher BMI, their cell proliferation was slower, and they had higher levels of DNA damage, recorded using all three assays.

Comparison of the ≥60 group with the 41–50 group demonstrated that the older subjects had highly significant values for BMI and hs troponin I. The 41–50 group had significantly higher values for Ca (mg), chlorides, iron (mg), energy kJ, carbohydrate maltose (g), sodium, phosphorous, carbohydrate sucrose, vit B1-thiamin, zinc, total fat, MUFA, SFA, alcoholic beverages, meat and meat products, sugars, and TG.

Comparison of the group ≥60 with ≤40 demonstrated that the older subjects had significantly higher values for carbohydrate maltose, SFA, hs troponin I, folate, fT4, HOMA-IR, M1, M3, M4, NDI, and MN. Group <40 had significantly higher levels of sugars.

Comparison of the group 41–50 with ≤40 demonstrated that the older subjects had significantly higher levels of niacin, proteins (g), vit B2-riboflavin (mg), vitB1, nitrogen (g), vit B12, zinc (mg), meat and meat products, and MN.

Comparison of the group 51–59 with ≤40 demonstrated significantly that the older subjects had higher levels for hs troponin I, TSH, BMR, M1, and MN. Group <40 had significantly higher levels of sugars (g), fT3, HOMA-IR, M2, M3, M4, and NDI.

A comparison of the 51–59 group with the 41–50 group demonstrated that the younger subjects had significantly higher levels of hs troponin I and M3.

When all three subgroups (51–59, 41–50, and ≤40) were inter-compared (Table 5), the 41–50 group consumed the least non-alcoholic beverages and more alcoholic beverages, along with the highest number of snacks and preserves. DII values were the highest in the ≤40 group. BMI, BMR, BFM, SMM, and weight were the highest in the 51–59 group. It is important to mention that this sub-group had the highest level of damage in terms of micronucleus frequency, apoptosis, and comet assay values. The highest NB and NPB frequencies were found in the 41–50 group.

To additionally prove the age effects on the parameters of interest, we further divided the entire group into age groups, >50 (n = 27) and ≤50 years (n = 26) (considering the already obtained results and considering that both groups had a similar number of individuals) (Appendix A). Obtained results suggest that the younger group (≤50 years) had a diet with a more diverse food group intake, while subjects older than 50 years had higher weight, BFM, BMI, and M1 values. They also had lower NDI values and higher DNA damage parameters measured by all 3 assays (MN, NB, NPB, apoptosis, necrosis, Fpg, and TI values).

Considering that by conducting all analyses, which are documented in the above-reported tables and Appendix A, we demonstrated that increases in BMR and BMI are connected with a decrease in the proliferation index, an increase in the frequency of mononuclear cells, and an increase in the DNA damage parameters in all three assays, we wanted to check how individual DNA damage parameters look (Figure 4 and Figure 5). We have previously established that for the Croatian population, the reference range for MN frequency goes up to 12.5 MN per 1000 binucleated cells [16]. This was also similar to the HUMN group reference range [14,15,39]. There were 13 individuals whose values were above that range. According to the HUMN group, the established NB reference range goes up to 5 NB per 1000 BN [14,15,39]. We had 21 individuals with levels of NB higher than the referent value. The same group established up to 10 NPB as a reference range, and we had 10 individuals exceeding those values [14,15,39].

As for oxidative DNA damage, there are no referent values, and the values were supposed to be around 0%, but on the group level the average was 6%, but we had 20 individuals exceeding those values. As for the comet assay, the existing literature from our group [17,18] established that healthy individuals have up to 9%, cancer-free individuals up to 12.4%, cancer cases have up to 17.7%, and deceased people have 18% of TI values in DNA damage. None of the subjects in our study group had levels higher than 18%. In the entire group, 33 individuals had levels higher than 9%, 22 levels higher than 12.4%, and 3 levels higher than 17.7%.

In Table 4, Spearman correlations demonstrated a significant connection between potato intake and weight and BFM, and eggs/egg dishes with weight and MN frequency, and less DNA damage in Fpg and TI with cereals, fish, fruit, and vegetable intake. When grouping individuals in the normal DNA damage level group and with higher damage in both the comet and MN assays, those with habits of eating eggs daily and all types of potatoes daily, 2–4 times weekly, or 5–6 times weekly, white rice 2–4 times weekly, and spirits and margarin 2–4 times weekly were the ones who demonstrated higher levels of DNA damage, but mostly for the comet assay. In our small group, there was no specific food type that would demonstrate if someone from our severely obese individuals would have this severe condition due to eating or higher DNA damage. This means that although potatoes as a vegetable can lower the final DII index, the amount and frequency of eating can have an impact on making the diet less healthy and less anti-inflammatory.

Next, we wanted to check what was common in individuals with higher DNA damage (see Figure 6) when the food intake did not demonstrate meaningful results. In the comet assay, individuals with more than 9% of TI had significantly higher levels of TG from biochemical parameters; none were common with anthropometrical ones, and with the micronucleus assay, they had in common significantly higher levels of apoptotic cells and the ones in the M1 phase, meaning that this amount of damage found in the comet assay also points to changes in the proliferation of the cells. For the micronucleus assay, we found that people with more than 13 micronuclei also had significantly elevated levels of NPB and urates and that they were all older people. As for NB, it demonstrated that people with more than 5 NB had significantly higher levels of HOMA-IR. Both assays did not demonstrate significant differences in DII, but it was slightly higher in the group with higher damage levels.

Considering that we had 8 individuals with former tumors and 25 individuals with a family history of tumors, we wanted to explore the level of DNA damage in both assays for those individuals (Table 6). The non-tumor group and tumor group did not significantly differ in the values for different types of DNA damage, and the same was true for the group with a family history of tumors and those without tumors in the family.

However, when the individuals with tumor history were checked, we noticed that most of them had higher values for TI (higher than the healthy population and even higher than the cancer-free population) with most of them being in the cancer-prone group for the comet assay (five of eight) (Table 7). For the micronucleus assay, half of them demonstrated higher values than 13 MN. For the group with a family history of cancer, it was the same but with higher numbers (15 of 25 were in the cancer-prone group, and 7 of 25 had higher MN values, 6 NPB values, and 12 NB values).

At last, we made multiple regression models connecting the DNA damage values found in all three assays with the analyzed biochemical and anthropometric parameters. These relationships were already explained in previous analyses; this is only a graphical presentation of the different positive and negative influences on the DNA damage predictions. These predictions will help in new investigations on a higher number of patients and also serve as the model which will be either confirmed or changed with the higher number of participants. Figure 7 represents a model for the TI parameter in the comet assay and the Fpg comet assay, and Figure 8 for the MN, NB, and NPB parameters in the MN *cytome* assay. In all of our models, the presence of previous tumors or a family history of tumors did not influence the model.

## 4. Discussion

The novelty of this study is that, although performed on a relatively small group of individuals (n = 53), for the first time, three different DNA damage assays (for oxidative DNA damage, DNA damage/repair, and chromosomal instability) were used in the same study on the same whole blood samples of severely obese individuals. The study demonstrated that all anthropometric parameters were elevated in the group and that they all correlated together, although BMI and BMR showed weak correlations, demonstrating again the need to use both of those parameters in assessing the severity of obesity with anthropometric measurements.

Although the anthropometric measurements did not strongly correlate with biochemical parameters in general, we were able to show a moderate correlation of insulin, HOMA-IR, and urate levels with weight, SMM, BMR, and BMI. This only demonstrated that both types of approaches, anthropometric and biochemical ones, are much needed in obesity and its severity assessment.

In terms of biochemical parameters, the lipid profile (TC, LDL-C, TG), glucose, HOMA-IR, hsCRP, and urates were found to be higher than the reference range of values among the majority of the participants in the group, together with a lower MCV and a higher neutrophil count among the other half of the group. Inorganic phosphates and hs troponin levels, although in reference range levels, in all three types of the parameters (anthropometric, biochemical, or DNA damage) demonstrated some influence (correlations).

Even though the FFQ frequency did not demonstrate that some types of consumed food will have an impact or clear influence on any of the anthropometric, biochemical, or elevated DNA damage parameters, the trend of eating some types of food daily, weekly, and monthly was noticed in groups with higher DNA damage than expected (tail intensity in the comet assay >9% and >12.4%, micronucleus frequency >13).

DII itself demonstrated a weak correlation with oxidative DNA damage measured using the Fpg comet assay, and in our modeling of TI prediction, DII, together with lipid profile parameters, urates, inorganic phosphates, MCV, and even waist-to-hip ratio, was among the parameters most strongly influencing the prediction of DNA damage in severely obese individuals. It was also seen that DII values were the highest in the ≤40 years age group. We demonstrated that increases in BMR, BFM, and BMI are connected with a decrease in the proliferation index, an increase in the frequency of mononuclear cells, and an increase in the DNA damage parameters in all three assays. Furthermore, inorganic phosphates and hs troponin levels, although in reference range levels, again were demonstrated to have some influence (correlations). BMI, BMR, BFM, SMM, and weight were the highest in the 51–59 years age group. This sub-group had the highest level of damage in terms of micronucleus frequency, apoptosis, and comet assay values. The highest NB and NPB frequencies were found in the 41–50 years age group.

DNA damage assays, especially the comet assay, demonstrated sensitivity in detecting individuals with higher genomic instability and, therefore, an increased risk of further developing obese comorbidities, such as cancer. Among the participants with previous tumors, or a family history of cancer, who did not show different results from other participants that were less suspected to have genomic instability, the assay demonstrated that it could track at least half of them in the cancer-prone group with the higher DNA damage, 

We also demonstrated that all three assays should be used together in the assessment of the severity of obesity and its comorbidities. Each assay demonstrated different types of damage that occur at different points in the life spans of severely obese individuals. In the future, both DNA damage and biochemical parameters should be combined with anthropometric ones for further monitoring of obesity. Since the study group consisted of a small number of individuals and the correlation was found to be weak or moderate, the experiment should be replicated, and the experimental design should include a larger number of participants in order to confirm our results and possibly provide a stronger correlation. Such a design could also result in weak correlations, demonstrating the need to use the previously mentioned three different parameters (anthropometric, biochemical, and DNA damage) in the assessment of obesity severity and revealing the DNA damage-prone individuals.

Although still scarce, knowledge on disease-causing obesity biomarkers holds promise to be used for a more refined diagnosis of obesity beyond anthropometric measurements, for more precise identification of persons at high risk of obesity/disease/genomic instability/cancer development, and in individual systems medicine/therapy. Combining biomarkers should also give us a better insight into the disease development and treatment approach.

There are reports characterizing BMI as not an accurate marker of obesity [45,46,47,48,49]. This is mostly due to the fact that fat accumulation in men is generally in the abdominal area and in women in the peripelvic area and thighs [45]. There is also a problem of low nutrient intake in people with very low BMI (WHO), which is not recognized if only BMI is used, but most of the opposing scientists agree that in severely obese individuals (BMI ≥ 35 kg m^−2^), BMI is still a good predictor of obesity and adipose-associated risks [46,47,48,49]. There are studies demonstrating that BMR can be a marker of metabolic health, independent of adiposity in normal-weight and overweight individuals [50], and independently of BMI being associated with greater mortality risk [51], and could identify subgroups with a greater risk of obesity comorbidities such as some types of cancer, that would not have otherwise been identified solely with BMI [52]. With the weak correlation found in our study, this fact only confirmed that both BMI and BMR should be used in the assessment of obesity severity and comorbidities.

The major cause of severe obesity is a long-term imbalance between energy intake and energy expenditure, which leads to weight and BMI gain and consequently an increase in all other anthropometric parameters [53,54]. Such an imbalance can be counteracted by a diet characterized by a low intake of high-energy-dense foods [e.g., sugar-sweetened beverages (SSBs), processed foods] and a high intake of low-energy-dense foods (e.g., fruit, vegetables, and whole-grain products) [55], meaning that high intake of the first group and low intake of the second food group can have a huge impact on obesity development, and there is plenty of evidence to prove that in the literature. However, a systematic review and dose–response meta-analysis of prospective studies found on PubMed and Web of Science until the end of 2018 demonstrated very low- to low-quality evidence that certain food groups have an impact on different measurements of adiposity risk [56] and our study found a similar pattern. It seems that in severely obese there are complicated and complex mechanisms of factors that contribute to the severity of obesity, and that the food, food habits, and food groups cannot be the only factors influencing that condition and that in severe obesity they are not a crucial factor anymore. Moreover, calculated DII from FFQ in our study demonstrated its weak connection with certain anti-inflammatory food groups’ intake such as cereals, fruit, or fish intake and it also demonstrated that with aging, individuals tend to eat less food but also use a higher DII diet (food). Although we cannot categorize severe obese risks according to the food groups, inflammatory diet type did show some influence and DII demonstrated its significant connection with measured oxidative DNA damage and in the model of DNA damage predictivity with the parameters used in this study.

At the same time, BMI and BMR were connected with a decrease in cell proliferation and DNA damage creation. We already mentioned in the introduction part that obesity is associated with an elevated risk of several NCDs, with evidence that DNA damage, mostly oxidative but also DNA damage in general, plays a key role in the development of most common NCDs, including cancer (even 13 different types), and that among other assays, comet, Fpg comet, and micronucleus *cytome* assays are sensitive and appropriate methods for DNA damage assessment but have not yet been fully explored in their potential for clinical use with other assessment biomarkers [12,13,19]. Our previous results implied that oxidative and DNA damage measured with the comet assay itself is not involved in the early stage of metabolic syndrome (MetS) and, therefore, cannot serve as an early marker of MetS [57,58]. Interestingly, BMI within the normal range did not influence DNA damage levels (comet assay) [59], but an increase in BMI, especially in severe obesity, lowered the activity of DNA repair mechanisms, especially nucleotide excision repair (NER), to the point of total inhibition, and also base excision repair (BER), which stayed active as the only repair mechanism, although with lower activity/capacity [26]. With the weight loss, activities of both repair mechanisms (maybe not in full but still!) can be restored [26]. Also, our previous cohort study focused on comet assay and NCDs demonstrated that age, smoking habit, and the presence of NCDs showed significantly increased mortality risks, but that from all of them in the model, the presence of NCD was the strongest predictor of mortality, but also that the case of chronic disease did not modify the association between DNA damage and mortality rates [18,19]. The results of that study also provided support for the hypothesis that an increased level of DNA damage represents a relevant event in the pathway leading to chronic disease and eventually death. In addition, the use of circulating leukocytes suggests that DNA damage can be suitably measured in this surrogate tissue to estimate mortality risk. The observation that the association between DNA damage and risk of death is not dependent on proximity to the outcome, i.e., is not driven by the possible presence of the disease, is consistent with the hypothesis that measuring DNA damage in healthy individuals and subjects with the disease at any time may predict NCD and death.

Although the comet assay, after an analysis of a huge registry/database of more than 19,000 individuals, has established certain values for healthy, cancer-free, cancer-prone, and deceased populations (the ones that will be in the mortality registry) [18,19], the micronucleus cytome assay has not been fully explored with all its parameters, usually only for MN frequency [21,25,27,60]. As can be seen in the mentioned reviews, although both assays demonstrated a positive connection between elevated MN frequency and comet assay parameters and obesity, studies and results are still scarce, with very little data on the severely obese without studies using both assays on the severely obese in those reviews [21,25,27,60]. Considering that NBs are a biomarker of gene amplification, NPBs are a biomarker of di-centric chromosomes and can demonstrate different types of DNA damage in comparison to the MN assay, which is a biomarker of whole/or partial chromosome loss. The changes in DNA damage were visible as the increase in NB and NPB frequencies in the 41–50 years, group but still with the normal proliferation of the cells in micronucleus assay. In the next group, 51–59 years old, with increasing BMI and BMR, a decrease in cell proliferation was observed but also an increase in micronucleus frequency.

From biochemical parameters, elevated levels of glucose demonstrated a connection with an increase in apoptotic and necrotic cells and cells with higher damage in the comet assay. Hs CRP by itself did not demonstrate a connection with DII or DNA damage, but it was found that a decrease in MCV was connected with elevated HOMA-IR and insulin levels, and platelet concentrations were connected with hsCRP. The results demonstrated an association between DNA damage levels and cell proliferation with anthropometric measurements and severity of obese status, together with inorganic phosphates, hs troponin I levels (although all within the reference range), and elevated levels of urates, insulin, and HOMA-IR. An increase in calcium levels, inorganic phosphates, and chlorides demonstrated a connection with comet assay DNA damage, necrosis, a decrease in cell proliferation, and an increase in MN frequency.

Severe obesity is associated with glucose intolerance, hyperinsulinemia, insulin resistance, hypertriglyceridemia, decreased high-density lipoprotein (HDL) cholesterol levels, and high blood pressure and can be a precursor of cardiovascular disease and type 2 diabetes mellitus (T2DM) [61]. Changes in biochemical blood parameters like in our study were seen also in other studies and demonstrated that severe obesity with its comorbidities is associated with increased chronic inflammation [62,63,64]; elevated levels of inflammatory markers, including C-reactive protein, ferritin, cytokines [65,66], total white blood cell (WBC) count [67,68,69], increased platelet count and activation [70,71], with elevated red blood cell (RBC) parameters, such as hemoglobin (Hb) and hematocrit, and electrolyte disorder [72], and also dysregulated dietary phosphate that is also connected with cancer risk [73]. Although a combination of multiple factors intervenes at different stages of obesity and in obesity-related genomic instabilities and cancer progression, it seems that obesity status can act on different fields by changing the hematological picture and hematopoietic stem cells’ quiescence and that low-grade chronic metabolic (oxidative) stress as a consequence can lead to DNA damage accumulation but not directly to mutation accumulation [74]. On the other side, first an increase in NBs and NPBs frequency (first age group), then a decrease in cell proliferation and later an increase in MN frequency (second and third age groups), oxidative DNA damage, and primary DNA damage measured with comet and a lowering of DNA repair capacities with the BMI increase also point out another direction: accumulated genomic instability that can further lead to cancer but also to higher mortality risk. In a multicohort study on more than 120,000 Europeans, Nyberg et al. (2018) demonstrated that severe obesity assessed with the BMI index (≥35 kg m^−2^) is connected with a significant reduction in the disease-free years (diseases connected with obesity, mostly NCD such as type 2 diabetes, coronary heart disease, stroke, cancer, asthma, and chronic obstructive pulmonary disease) of 7 to 8 (up to 10) years in both males and females, smokers and non-smokers, physically active, and inactive, and across the social hierarchy, in people between 40 and 75 years of age who did not have any disease when first entering the study [10]. These results only point out the necessity of further biomarkers exploring in detecting more sensitive individuals and creating new approaches to the studying of obesity processes and new individual therapy approaches.

Further steps should include study replication with a larger number of participants, the development of a registry of obese patients, which would simplify the monitoring and follow-up processes, and connecting that registry with the mortality registry. In addition, individuals should be educated on how to eat healthy and given appropriate diets, explanations about the type of food with proinflammatory or anti-inflammatory potential, and how to use that information. All of these suggestions could be performed under medical surveillance and should also include BER and NER comet assays in order to check whether DNA damage has been reduced after the diets and whether repair mechanisms have gained their activity again. All significant biomarkers found in this study should be regularly used in the systemic and individual approaches to each obese and especially severely obese patient. New data on those biomarkers would also give the possibility for new data meta-analysis and provide new guidelines for future treatment and prevention of severe obesity and cancer, and maybe obesity in general.

## Figures and Tables

**Figure 1 nutrients-15-00899-f001:**
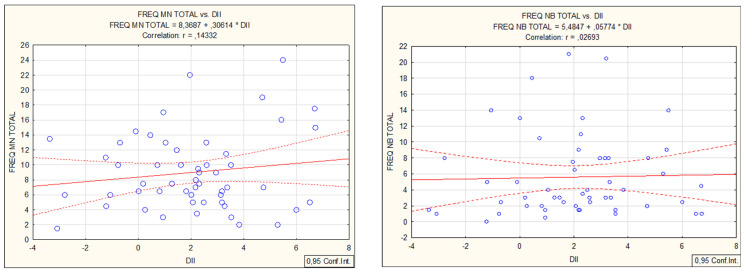
Spearman correlations of MN—micronuclei, NB—nuclear buds, NPB—nucleoplasmic bridges and apoptosis in micronucleus *cytome* assay with inflammatory food index DII.

**Figure 2 nutrients-15-00899-f002:**
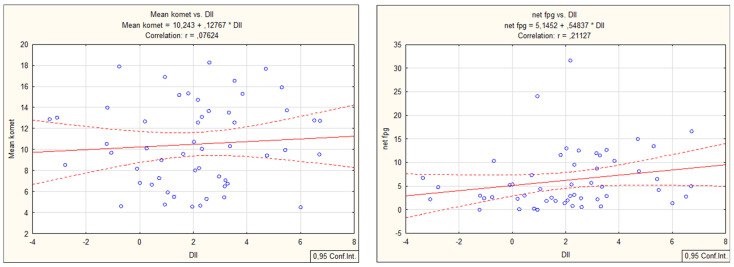
Spearman correlations of TI (%DNA in comet tail) in comet assay (no significant correlation) and Fpg comet assay with inflammatory food index DII.

**Figure 3 nutrients-15-00899-f003:**
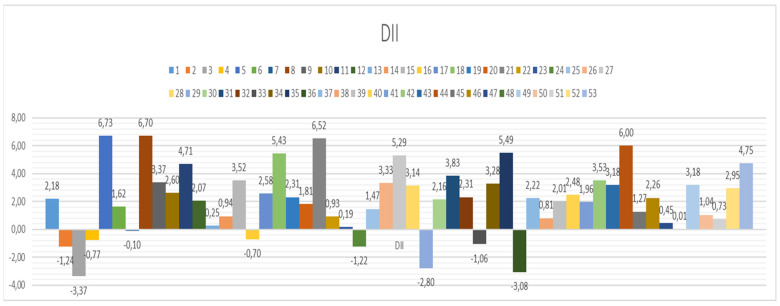
Calculated dietary inflammatory index (DII) on the individual level for all 53 individuals included in the study.

**Figure 4 nutrients-15-00899-f004:**
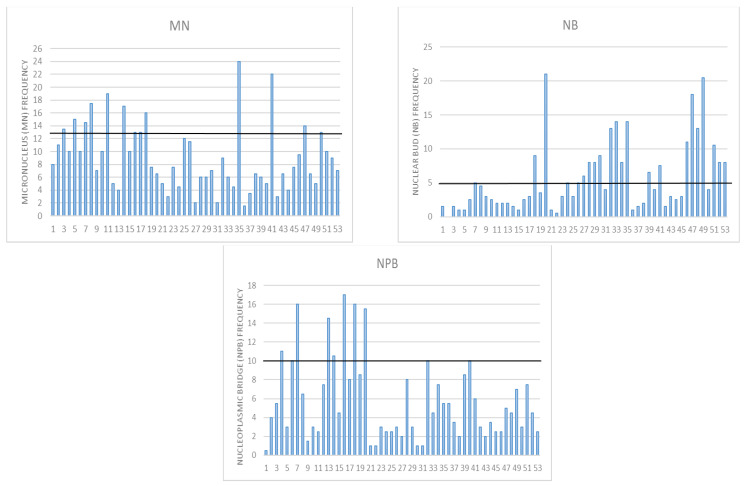
Individual DNA damage for micronucleus *cytome* assay parameters: MN-micronucleus with a reference range up to 12.5 per 1000 binucleated BN cells; NB—nuclear buds with a reference range up to 5 per 1000 BN; NPB—nucleoplasmic bridges with reference range up to 10 per 1000 BN cells in a healthy human population.

**Figure 5 nutrients-15-00899-f005:**
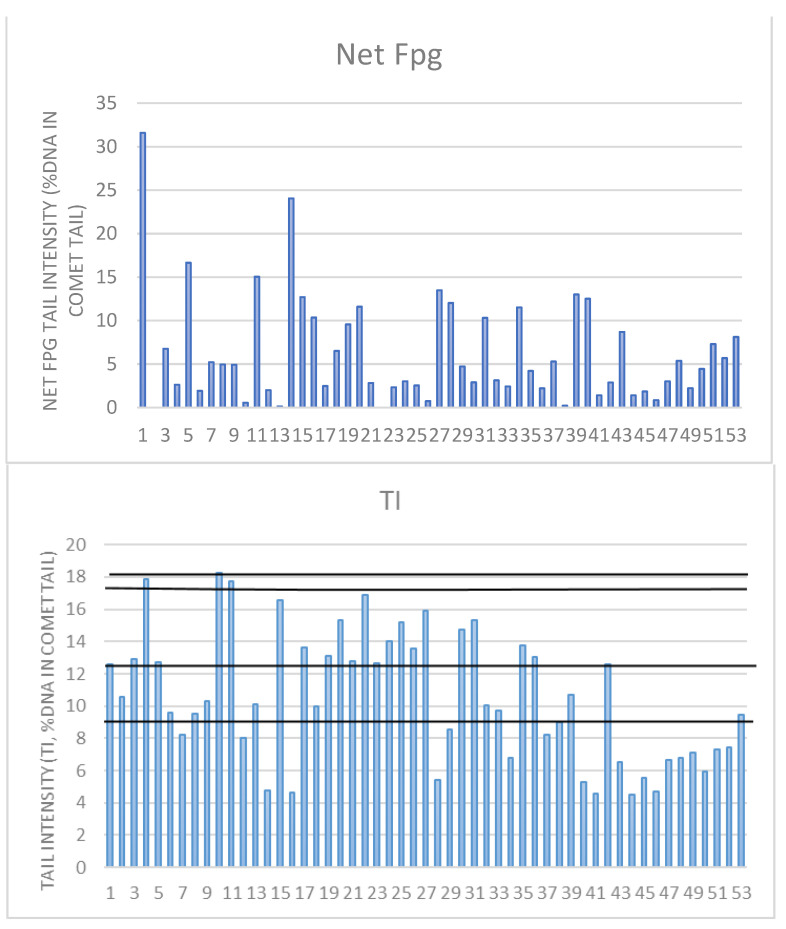
Individual DNA damage for Fpg and alkaline comet assay TI parameters. For Net Fpg TI, there is no reference value for oxidative damage; the mean value for the entire group was 6%. We demonstrated before that alive and healthy individuals have up to 9% of TI, cancer-free up to 12.4%, cancer cases 17.7%, and deceased 18%. The highest value in this study group was below 18% (tail intensity).

**Figure 6 nutrients-15-00899-f006:**
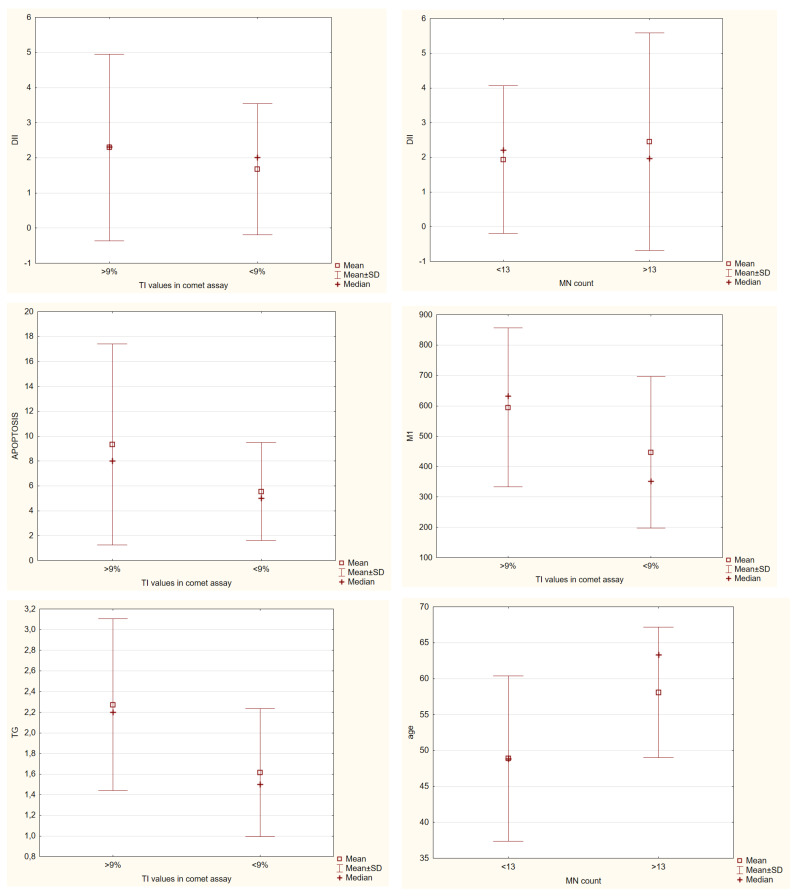
Significant differences in biochemical parameters among groups with higher or lower DNA damage parameters in both comet and micronucleus *cytome* assay, Mann–Whitney U test, *p* ≤ 0.05.

**Figure 7 nutrients-15-00899-f007:**
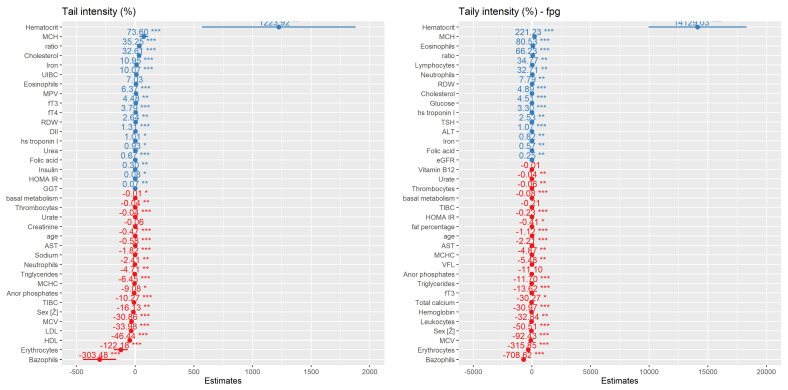
Stepwise regression model for DNA damage prediction of tail intensity (TI, %) values in alkaline comet and Fpg comet assay for obese group of people with BMI ≥ 35 kg m^−2^ based on the biochemical and anthropometric parameters used in the study. Values are categorized by coefficient estimates for TI value prediction, while *p* values reflect the importance of each variable for model prediction. Negative values have negative correlation (red) and positive values have positive correlation with TI (blue) *—*p*-value < 0.05, **—*p*-value < 0.01, ***—*p*-value < 0.001.

**Figure 8 nutrients-15-00899-f008:**
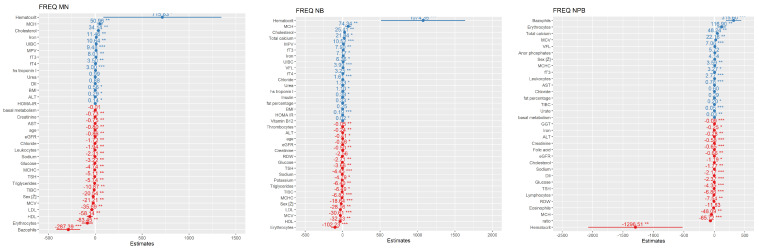
Stepwise regression model for DNA damage prediction of micronucleus (MN), nuclear bud (NB) and nucleoplasmic bridge (NPB) frequency in MN *cytome* assay for obese group of people with BMI ≥35 kg m^−2^ based on the biochemical and anthropometric parameters used in the study. Values are categorized by coefficient estimates for DNA damage prediction, while *p* values are responding to the importance of each variable for model prediction. Negative values have negative correlation (red) and positive values have positive correlation with TI (blue) *—*p*-value < 0.05, **—*p*-value < 0.01, ***—*p*-value < 0.001.

**Table 1 nutrients-15-00899-t001:** Anthropometric parameters of the entire group (n = 53) with BMI ≥ 35 kgm^−2.^

Variable/Unit, Reference Values	Mean	Median	Min	Max	SD	SE	Spearman Correlation (R)
Weight/kg, 85–115	126.62	126.00	95.30	183.60	19.29	2.65	**SMM(0.84), BFM(0.68), BMI(0.72), BMR(0.83)**
SMM/kg, 19.8–24.2	37.27	34.90	23.80	56.20	8.13	1.12	**weight(0.84),** BMI (0.34), **BMR(1), PBF%(−0.47), VFL(−0.42)**
BFM/kg, 10.8–17.2	60.78	62.00	40.40	90.30	11.52	1.58	**weight(0.68), PBF%(0.56), BMI(0.88),** VFL(0.28)
PBF/%, 10–20 M, 18–28 F	47.94	50.00	30.30	64.50	6.59	0.91	**BFM(0.56), BMI(0.37), VFL(0.37), SMM(−0.47), BMR(−0.48)**
BMI/kgm^−2^, 18.5–25	44.58	43.60	35.40	57.50	5.77	0.79	**weight(0.72),** SMM(0.34), **BFM(0.88), PBF%(0.37),** BMR(0.33)
BMR/kcal day^−1^, 1255–1451	1792.28	1703.00	1299.00	2462.00	290.91	39.96	**weight(0.83), SMM(1),** BMI(0.33), **PBF%(−0.48), VFL(−0.42)**
W-H ratio, 0.75–0.85	1.05	1.06	0.70	1.27	0.11	0.01	**-**
VFL, 10	19.77	20.00	16.00	20.00	0.80	0.11	BFM(0.28), **PBF%(0.37), SMM(−0.42), BMR(−0.42)**

F—female, M—male, SMM, BFM, PBF, BMI, BMR, VFL SMM—skeletal muscle mass in kg, BFM—body fat mass in kg, BMI—body mass index, PBF—the percentage of body fat in %, W-H ratio—waist-hip ratio, VFL—visceral fat level, BMR—basal metabolic rate; all measured with InBody 270. In bold are moderately (0.4–0.59), strongly (0.6–0.79), and very strongly (0.8–1) correlated variables.

**Table 2 nutrients-15-00899-t002:** Biochemical parameters measured in the entire group (n = 53) with BMI ≥ 35 kg m^−2^ with Spearman correlation.

Variables, Reference Values	Mean	Median	Min	Max	SD	SE	Spearman Correlations
Erythrocytes, 3.86–5.08 F, 4.34–5.72 M ∗ 10^12^/L	4.85	4.82	3.86	5.81	0.41	0.06	
Hemoglobin, 119–157 F, 138–175 M g/L	139.04	138.00	107.00	165.00	13.65	1.88	
Hematocrit, 0.356–0.47 F, 0.414–0.53 M L/L	0.41	0.41	0.33	0.48	0.03	0.00	
MCV, 83–97.2 FM fl	84.63	85.30	62.80	94.30	6.14	0.84	insulin(−0.56), HOMA-IR(−0.51), erythrocytes(−0.46)
RDW %, 9–15 FM	13.91	13.50	12.40	**19.70**	1.28	0.18	chlorides(0.42)
Platelets, 158–424 FM ∗ 10^9^/L	272.30	249.00	167.00	**563.00**	86.76	11.92	leukocytes(0.61), neutrophils(0.58), monocytes(0.41)
Leukocytes, 3.4–9.7 FM ∗ 10^9^/L	8.20	7.70	5.04	**14.26**	2.12	0.29	neutrophils(0.91), monocytes(0.71), platelets(0.61), lymphocytes(0.52), %neutro(0.44), mono%(−0.43)
Basophils, 0–0.06 FM, ∗ 10^9^/L	0.03	0.03	0.01	0.09	0.02	0.00	
Basophils %, 0–1 FM	0.44	0.40	0.10	**1.30**	0.24	0.03	
Neutrophils, 2.06–6.49 FM ∗ 10^9^/L	4.85	4.63	2.00	**9.65**	1.66	0.23	leukocytes(0.91), lymphocytes%(−0.65), monocytes(0.63), platelets(0.58), hsCRP(0.45), monocytes%(−0.42)
Neutrophils %, 44–72 FM	58.28	58.30	38.60	**75.00**	8.22	1.13	lymphocytes%(−0.95), lymphocytes(−0.45), leukocytes(0.44), hsCRP(0.44)
Lymphocytes, 1.19–3.35 FM *10^9^/L	2.39	2.26	1.18	**4.10**	0.71	0.10	leukocytes(0.52), LDL-C(0.47), neutrophils%(−0.45), cholesterol(0.43)
Lymphocytes %, 20–46 FM	29.77	29.00	14.60	**49.20**	7.88	1.08	neutrophils(−0.65), hsCRP(−0.43)
Monocytes, 0.12–0.84 FM ∗ 10^9^/L	0.68	0.64	0.36	**1.24**	0.18	0.02	leukocytes(0.71), neutrophils(0.63), platelets(0.41), W-H-ratio(0.40)
Monocytes %, 2–12 FM	8.48	8.20	4.80	**13.10**	1.81	0.25	leukocytes(−0.43), neutrophils(−0.42), urates(0.41
Glucose, 4.4–6.4 FM mmol/L	**7.18**	**6.40**	**4.30**	**19.10**	**2.30**	**0.32**	HOMA-IR(0.57), hs troponin(0.50), fT4(0.49)
Urate, 134–337 F, 182–403 M µmol/L	**412.19**	**395.00**	**247.00**	**838.00**	**122.28**	**16.80**	monocytes%(0.41), platelets(−0.39), SMM(0.37), BMR(0.37), hs troponin(0.35)
Urea, 2.8–8.3 mmol/L	6.64	6.00	3.90	**18.70**	2.58	0.35	erythrocytes(0.50), hemoglobin(0.45), hematocrit(0.41)
Total calcium, 2.14–2.53 FM mmol/L	2.43	2.43	2.23	**2.71**	0.11	0.02	inorganic phosphates(0.51), chlorides(0.46), albumin(0.41)
Inorganic phosphates, 0.79–1.42 FM mmol/L	1.03	0.93	0.74	**1.53**	0.23	0.03	chlorides(0.71), Ca total(0.51), hstroponin(0.51), weight(0.36), BMR(0.36), platelets(−0.36), fT4(0.35), SMM(0.35), BMI(0.35)
Chlorides, 97–108 mmol/L	50.55	1.35	0.84	108.00	50.91	6.99	inorganic phosphates(0.71), Ca total(0.46), BMI(0.46), hs troponin(0.44), RDW(0.42), weight(0.41), platelets(−0.39), BMR(0.38), SMM(0.37)
Albumin, 40.6–51.4 FM g/L	44.07	44.00	38.20	49.30	2.48	0.34	BFM(−0.47),Ca total(0.41), fT3(0.41)
hsCRP, ≤5 FM mg/L	**9.94**	**6.78**	**0.99**	**39.64**	**9.20**	**1.26**	lymphocytes%(−0.43)
hs troponin I, ≤15.6 F, ≤34.2 M ng/L	3.12	2.00	0.10	18.70	3.47	0.48	inorganic phosphates(0.51), glucose(0.50), BMR(0.49), SMM(0.46), chlorides(0.44), VFL(−0.39), basophils(0.37), weight(0.37), basophils%(0.36), TC(−0.36), fT4(0.36), LDL-C(−0.36), urates(0.35)
TC, ≤5 FM, mmol/L	**5.65**	**5.60**	**3.00**	**10.10**	**1.46**	**0.20**	LDL-C(0.96), lymphocytes(0.43), TG(0.41), HDL-C(0.40), hs troponin(−0.36)
HDL-C, ≤1.2 F, ≤1 M mmol/L	1.24	1.20	0.80	2.10	0.30	0.04	BMR(−0.45), SMM(−0.44), weight(−0.41), TC(0.40)
LDL-C, ≤3 FM mmol/L	**3.56**	**3.20**	**1.00**	**7.70**	**1.25**	**0.17**	TC(0.96), lymphocytes(0.47), hs troponin(−0.36), TG(0.35)
TG, ≤1.7 FM mmol/L	**2.02**	**2.00**	**0.80**	**4.60**	**0.82**	**0.11**	TC(0.41), LDL-C(0.35)
TSH, 0.35–4.94 FM mIU/L	2.38	2.05	0.08	**7.69**	1.39	0.19	
fT4, 9–19.5 FM pmol/L	12.80	12.60	9.32	18.57	1.91	0.26	glucose(0.49)
fT3, 2.42–6 FM, pmol/L	4.22	4.25	2.64	5.59	0.72	0.10	albumin(0.41), BMI(−0.40)
Insulin, 2–22 FM mU/L	17.40	15.10	4.30	**56.30**	8.87	1.22	HOMA-IR(0.90), MCV(−0.56), weight(0.40), erythrocytes(0.39)
HOMA-IR, <2.5 FM	**5.85**	**4.68**	**1.40**	**19.77**	**4.06**	**0.56**	insulin(0.90), glucose(0.57), MCV(−0.51), BMI(0.43), weight(0.41)

F—female, M—male, MCV—mean corpuscular volume of red blood cells, RDW—red cell distribution width, hsCRP—high sensitivity C-reactive protein, TC—total cholesterol; in bold are levels higher than the reference range values. Spearman correlation expressed are the ones that were moderately (0.4–0.59), strongly (0.6–0.79), and very strongly (0.8–1) correlated variables.

**Table 3 nutrients-15-00899-t003:** DNA damage parameters measured in all three assays on the group level (n = 53) with Spearman correlation (SC).

Micronucleus Assay, Reference Range	Mean	Median	Min	Max	SD	SE
M1	538.96	545.00	186.00	912.00	264.62	36.35
SC: NDI(−0.98), M4(0.91), M3(−0.91), M2(−0.89), NB(−0.43), inorganic phosphates(0.52), chlorides(0.50), BMI(0.48), fT3(−0.45), weight(0.38), Ca total(0.38), BFM(0.36), MN(0.35)						
M2	316.94	330.00	84.00	575.00	156.10	21.44
SC: M1(0.89), NDI(0.81), M4(0.72), M3(0.71), fT3(0.49), chlorides(−0.48), NB(0.45), BMI(−0.46), BFM(−0.38), inorganic phosphates(−0.42), weight(−0.35), MN(−0.35)						
M3	51.83	51.00	1.00	140.00	42.55	5.84
SC: NDI(0.93), M4(0.92), M1(−0.91), M2(0.71), inorganic phosphates(−0.55), chlorides(−0.53), BMI(−0.46), hs troponin(−0.43), fT3(0.42), weight(−0.40), Ca total(−0.40), MN(−0.39), fT4(−0.38), NB(0.35)						
M4	91.87	61.00	0.00	302.00	94.59	12.99
SC: NDI(0.96), M3(0.92), M1(−0.91), M2(0.72), chlorides(−0.58), inorganic phosphates(−0.55), BMI(−0.54), NB(0.47), weight(−0.46), hs troponin(−0.43), Ca total(−0.41), fT3(0.40), SMM(−0.35), BFM(−0.35)						
NDI, ~2	1.70	1.74	1.09	2.55	0.48	0.07
SC: M1(−0.98), M4(0.96), M3(0.93), M2(0.81), inorganic phosphates(−0.53), chlorides(−0.52), BMI(−0.48), weight(−0.42), fT3(0.41), NB(0.40), hs troponin(−0.39), Ca total(−0.37)						
FREQ MN, 0–12.5	9.00	7.50	1.50	24.00	5.10	0.70
SC: inorganic phosphates(0.42), M3(−0.39), fT3(−0.39)						
FREQ NB, 0–5	5.60	3.50	0.00	21.00	5.12	0.70
SC: M4(0.47), chlorides(−0.46), M2(0.45), Ca total(−0.43), M1(−0.43), NDI(0.40), BMI(−0.39), fT3(0.39), W-H-ratio(0.38), M3(0.35)						
FREQ NPB, 0–10	5.74	4.50	0.50	17.00	4.31	0.59
SC: urates(0.36)						
APOPTOSIS, 0–7%	7.91	7.00	0.00	42.00	7.02	0.96
SC: mean comet(0.36)						
NECROSIS, 0–9%	2.79	0.00	0.00	33.00	7.45	1.02
SC: TSH(0.53), inorganic phosphates(0.37), glucose(0.35)						
Alkaline comet assay						
Mean comet (TI), 0–9%	10.51	10.04	4.50	18.27	4.00	0.55
SC: TG(0.43), chlorides(0.41), apoptosis(0.36), glucose(0.35)						
Fpg alkaline comet assay						
Net Fpg (TI), 0%	6.28	4.45	0.00	31.59	6.20	0.85

SD—standard deviation, SE—standard error, MN—micronuclei, NB—nuclear buds, NPB—nucleoplasmic bridges. TI—tail intensity (%DNA in comet tail), Net Fpg—TI% of oxidative DNA damage. Spearman correlation expressed are the ones that were moderately (0.4–0.59), strongly (0.6–0.79), and very strongly (0.8–1) correlated variables.

**Table 4 nutrients-15-00899-t004:** Spearman’s correlation coefficient between different food groups derived by food frequency questionnaire and FETA program and anthropometric, biochemical, or DNA parameters (n = 53).

Food Groups (g)	Mean	Median	Min	Max	SD	SE	Spearman Correlation(R)
Alcoholic beverages	27.26	7.00	0.00	316.00	52.57	7.22	Positive: Alcohol(0.95), Negative: Age(−0.31)
Cereals and cereal products	264.80	188.91	0.56	2585.28	364.44	50.06	Negative: DII(−0.44), TG(−0.27), Net Fpg(−0.36), TI (−0.28)
Eggs and egg dishes	16.28	17.50	0.00	50.00	13.07	1.80	Positive: weight(0.28), MN(0.33)
Fats and oils	12.32	6.51	0.00	177.12	26.51	3.64	Negative: TG(−0.29)
Fish and fish products	27.45	24.15	0.00	101.00	19.14	2.63	Negative: DII(−0.32), Net Fpg(−0.30)
Fruit	184.25	129.75	17.50	1477.80	232.51	31.94	Negative: DII(−0.47)
Meat and meat products	158.88	134.89	20.65	613.13	108.22	14.87	Negative: DII(−0.47), M4(−0.28)
Milk and milk products	101.37	78.30	0.00	423.20	97.54	13.40	Positive: Insulin(0.29), Negative: DII(−0.38)
Non-alcoholic beverages	634.95	515.00	9.48	1791.10	427.55	58.73	Negative: DII(−0.35)
Nuts and seeds	13.40	2.10	0.00	137.66	26.01	3.57	-
Potatoes	91.49	74.54	0.00	392.69	75.24	10.34	Positive: Weight(0.33), BFM(0.28), Negative: DII(−0.46)
Soups and sauces	97.30	78.00	0.00	377.90	76.06	10.45	Positive: NPB(0.28), Negative: DII(−0.34)
Sugars	50.50	22.49	0.00	530.50	83.34	11.45	Negative: hs troponin(−0.35), TI(−0.34), age(−0.51)
Preserves and snacks	225.85	213.05	20.64	747.81	140.21	19.26	Positive: HDL-C(0.28), Negative: DII(−0.64), Net Fpg(−0.27)
Vegetables	4.34	5.00	0.00	5.00	1.71	0.23	Positive: fT3(0.31), M2(0.37), M3(0.40), M4(0.35), NDI(0.39), NB(0.44), Negative: hs troponin(−0.32), weight(−0.34), M1(−0.38), TI(−0.29)

DII—dietary inflammatory index, TG—triglycerides, TI—tail intensity (%DNA in comet tail), MN—micronuclei, Net Fpg—TI of oxidative DNA damage, M4—tetra nucleated cells, BFM—body fat mass, NPB—nucleoplasmic bridges, M2—binucleated cells, M3—trinucleated cells, NDI—nuclear proliferation index, NB—nuclear buds.

**Table 5 nutrients-15-00899-t005:** The differences in the nutrient intake and food group intake (based on the FFQ questionnaire), together with anthropometric, biochemical and DNA damage parameters observed after dividing the group by age into subgroups: <60 years, 51–59 years, 41–50 years, and ≤40 years, with higher levels in bold.

	Age 51–59, n = 9						Age 41–50, n = 17						Age < 40, n = 9					
Variable	Mean	Median	Min	Max	SD	SE	Mean	Median	Min	Max	SD	SE	Mean	Median	Min	Max	SD	SE
Alpha carotene Alcohol (mcg)	105.04	101.09	8.58	203.51	76.84	25.61	**204.76**	**106.30**	**4.62**	**994.71**	**261.21**	**63.35**	123.85	100.32	9.43	201.53	72.72	24.24
Alcohol (g)	5.70	0.51	0.00	34.79	11.33	3.78	**5.93**	**2.50**	**0.00**	**34.03**	**9.82**	**2.38**	5.60	0.51	0.00	43.75	14.36	4.79
Beta carotene (mcg)	1573.50	1634.15	608.99	2479.17	690.09	230.03	**2218.75**	**1764.11**	**632.18**	**5381.19**	**1501.02**	**364.05**	1697.99	1125.44	301.25	4972.47	1492.09	497.36
Calcium (mg)	573.15	487.27	219.54	1238.94	294.13	98.04	**832.65**	**571.35**	**415.44**	**3734.16**	**795.68**	**192.98**	645.69	568.80	263.16	1184.92	329.03	109.68
Carotene—total (carotene equivalents) (mcg)	1845.98	2056.93	716.80	2729.51	802.05	267.35	**2598.26**	**2023.78**	**638.77**	**5945.25**	**1705.91**	**413.74**	1947.99	1277.80	359.58	5597.43	1697.27	565.76
Carbohydrate—total (g)	185.08	179.68	105.96	298.98	68.85	22.95	**300.65**	**214.45**	**54.23**	**1771.66**	**390.31**	**94.66**	250.55	175.76	104.99	633.13	176.59	58.86
Cholesterol (mg)	286.21	305.22	118.47	460.33	114.99	38.33	**411.87**	**265.58**	**187.77**	**2284.20**	**491.91**	**119.31**	281.40	210.48	124.05	617.10	171.84	57.28
Chloride (mg)	3626.55	3867.86	1727.77	5274.23	1236.18	412.06	**4781.56**	**3537.11**	**2382.51**	**18,279.1**	**3747.57**	**908.92**	3585.57	2563.62	1528.34	7917.96	2386.75	795.584
Copper (mg)	1.53	1.36	0.76	3.43	0.80	0.27	**2.35**	**1.40**	**0.70**	**11.89**	**2.72**	**0.66**	1.24	0.97	0.39	3.10	0.85	0.28
Englyst Fibre—non-starch polysaccharides(NSP)(g)	13.71	14.06	7.91	18.94	3.94	1.31	20.05	15.74	7.85	62.67	13.89	3.37	**4677.40**	**10.94**	**3.81**	**42,003.0**	**13,997.1**	**4665.70**
Iron (mg)	10.14	11.46	5.01	14.39	2.87	0.96	**14.76**	**11.65**	**6.93**	**54.90**	**11.25**	**2.729**	9.61	7.84	3.71	20.95	5.78	1.93
Total folate (mcg)	241.21	253.96	121.49	342.18	82.86	27.62	**300.43**	**277.25**	**152.30**	**726.66**	**153.91**	**37.33**	203.91	175.71	73.20	432.60	125.17	41.72
Carbohydrate—fructose (g)	23.16	24.00	10.60	46.92	11.47	3.82	**25.66**	**16.24**	**5.48**	**110.14**	**25.85**	**6.27**	21.02	11.22	4.82	88.20	26.56	8.85
Carbohydrate—galactose (g)	0.54	0.27	0.00	1.44	0.55	0.18	**0.91**	**0.48**	**0.00**	**4.41**	**1.22**	**0.30**	0.32	0.12	0.00	1.01	0.34	0.12
Carbohydrate—glucose (g)	22.15	20.41	11.37	40.93	10.23	3.41	**26.48**	**15.62**	**4.90**	**125.52**	**28.60**	**6.94**	22.37	15.68	5.36	77.32	22.38	7.46
Carbohydrate—glucose (mcg)	68.18	69.99	31.57	104.29	24.76	8.25	**127.15**	**76.06**	**45.18**	**662.59**	**146.71**	**35.58**	85.85	62.65	38.10	194.69	53.28	17.76
Potassium (mg)	2750.84	2580.86	1869.45	3835.49	684.62	228.21	**3985.89**	**3824.90**	**1821.53**	**12,996.9**	**2649.54**	**642.61**	3002.97	2198.26	1155.01	7816.25	2120.70	706.90
Energy_kcal	1470.51	1511.38	844.57	2055.98	478.51	159.50	**2443.35**	**1709.93**	**1156.04**	**12,727.6**	**2728.69**	**661.81**	2132.80	2016.03	870.56	4563.85	1161.46	387.16
Energy_kj	6191.10	6358.97	3563.39	8645.21	2013.56	671.19	**10,273.7**	**7195.29**	**4805.64**	**53,529.7**	**11,478.7**	**2783.99**	8098.61	5961.07	3664.38	19,180.3	5060.99	1687.00
Carbohydrate—lactose (g)	3.66	3.14	1.44	8.61	2.25	0.75	**6.61**	**3.54**	**0.96**	**45.08**	**10.62**	**2.58**	5.51	5.28	1.18	10.34	2.98	0.99
Carbohydrate—maltose (g)	1.57	1.23	0.14	3.38	1.07	0.36	3.35	1.617	0.166	24.30	5.66	1.37	**3.85**	**2.54**	**0.84**	**11.90**	**3.53**	**1.18**
Magnesium (mg)	239.46	248.76	132.02	340.09	58.93	19.64	**353.51**	**283.81**	**146.11**	**1139.36**	**238.08**	**57.74**	254.28	186.78	100.45	687.19	183.24	61.08
Manganese (mg)	3.01	2.82	1.43	4.37	0.95	0.32	**4.00**	**2.785**	**1.39**	**14.70**	**3.35**	**0.81**	3.21	2.50	0.71	11.01	3.07	1.02
Sodium (mg)	2488.07	2642.56	1178.08	3557.61	863.31	287.77	**3376.43**	**2492.29**	**1631.38**	**13,993.4**	**2897.73**	**702.80**	2388.65	1677.65	1106.84	5050.05	1536.96	512.32
Niacin (mg)	21.88	24.37	13.55	27.37	5.14	1.71	**31.39**	**25.08**	**12.45**	**81.92**	**17.30**	**4.20**	18.68	14.39	9.13	39.30	10.06	3.35
Phosphorus (mg)	1097.58	1109.68	560.59	1730.58	347.45	115.82	**1667.92**	**1158.57**	**800.31**	**6624.44**	**1375.70**	**333.66**	1051.78	859.97	528.78	1872.52	504.47	168.16
Protein (g)	75.09	75.34	44.90	117.93	23.66	7.89	**111.66**	**85.30**	**53.46**	**363.30**	**74.73**	**18.12**	68.26	55.70	38.75	115.37	29.67	9.89
Vitamin A—retinol (mcg)	1427.28	926.11	168.39	5250.17	1522.96	507.65	**2065.08**	**1076.95**	**54.90**	**14,070.9**	**3473.56**	**842.46**	698.10	292.58	176.59	1738.48	694.60	231.53
Vitamin A—retinol equivalents (mcg)	1736.80	1381.10	511.82	5370.13	1450.29	483.43	**2501.95**	**1674.59**	**161.84**	**14,829.7**	**3546.31**	**860.11**	1022.37	553.03	253.20	2672.78	891.64	297.21
Vitamin B2—riboflavin (mg)	1.44	1.44	0.71	2.23	0.50	0.17	**1.99**	**1.22**	**0.86**	**8.31**	**1.80**	**0.44**	1.10	0.82	0.61	2.09	0.56	0.19
Selenium (mcg)	70.61	78.64	31.36	91.94	20.04	6.68	**85.26**	**73.02**	**37.65**	**263.07**	**54.23**	**13.15**	56.48	49.12	29.95	104.80	28.18	9.39
Carbohydrate—starch (g)	89.89	87.31	40.61	143.99	33.11	11.04	**150.15**	**123.03**	**13.84**	**800.47**	**174.38**	**42.29**	113.25	87.67	42.84	221.95	60.84	20.28
Carbohydrate—sucrose (g)	32.94	29.84	12.11	62.12	15.25	5.08	**70.33**	**31.15**	**12.13**	**605.67**	**139.84**	**33.92**	76.47	39.35	19.77	219.10	74.32	24.78
Vitamin B1—thiamin (mg)	1.27	1.16	0.74	1.95	0.41	0.14	**142.55**	**1.69**	**0.76**	**2395.00**	**580.44**	**140.78**	1.12	0.87	0.47	2.13	0.59	0.20
Nitrogen (g)	12.14	12.17	7.23	19.00	3.82	1.28	**18.18**	**13.76**	**8.62**	**59.56**	**12.26**	**2.97**	11.11	9.09	6.36	19.16	4.88	1.63
Carbohydrate—sugars (total) (g)	88.30	75.76	46.22	143.32	37.07	12.36	**141.13**	**78.11**	**33.95**	**939.79**	**212.75**	**51.60**	132.48	81.78	36.00	404.63	122.91	40.97
Vitamin B12—cobalamin (mcg)	7.38	7.07	1.45	19.61	5.39	1.80	**10.60**	**6.318**	**1.71**	**52.37**	**12.09**	**2.93**	4.38	2.98	1.65	8.41	2.64	0.88
Vitamin B6—pyridoxine (mg)	1.60	1.49	1.02	2.31	0.44	0.15	**2.40**	**1.80**	**1.08**	**7.14**	**1.49**	**0.36**	1.63	1.39	0.66	3.89	1.02	0.34
Vitamin C—ascorbic acid (mg)	86.22	76.94	45.63	147.31	36.58	12.19	**164.62**	**77.91**	**53.70**	**1111.13**	**254.00**	**61.60**	83.55	49.14	18.41	289.33	88.00	29.33
Vitamin D—ergocalciferol (mcg)	2.33	2.60	0.81	3.81	1.17	0.39	**3.69**	**2.11**	**1.35**	**22.85**	**5.07**	**1.23**	3.17	2.28	1.40	9.05	2.40	0.80
Vitamin E—alpha tocopherol equivalents (mg)	7.73	7.84	3.95	10.83	2.43	0.81	**14.10**	**9.99**	**4.21**	**75.19**	**16.19**	**3.93**	10.90	7.42	3.34	33.66	9.61	3.20
Zinc (mg)	8.37	8.28	4.51	14.26	2.77	0.92	**12.52**	**9.25**	**6.01**	**40.46**	**8.28**	**2.00**	7.35	5.94	3.75	12.70	3.38	1.13
Fat—total (g)	48.55	45.72	22.78	86.31	20.41	6.80	**91.93**	**63.09**	**29.39**	**513.83**	**112.25**	**27.23**	74.90	56.78	32.22	191.07	50.06	16.69
Monounsaturated fatty acids (MUFA—total) (g)	17.49	15.87	8.11	28.68	7.29	2.43	**36.54**	**23.69**	**8.86**	**210.71**	**46.88**	**11.37**	30.15	22.04	13.06	87.17	23.44	7.81
Polyunsaturated fatty acids (PUFA—total) (g)	8.32	7.89	4.29	13.60	3.16	1.05	**14.63**	**9.65**	**6.08**	**68.10**	**14.63**	**3.55**	10.29	6.90	3.60	32.18	8.78	2.93
Saturated fatty acids (SFA—total) (g)	17.75	16.40	7.57	39.98	9.58	3.19	**32.17**	**19.88**	**8.57**	**186.60**	**40.75**	**9.88**	27.59	24.29	10.79	54.82	14.42	4.81
Alcoholic beverages (g)	20.40	3.50	0.00	125.50	40.51	13.50	**52.01**	**35.63**	**0.00**	**316.00**	**76.18**	**18.48**	27.37	3.50	0.00	138.00	46.22	15.41
Cereals and cereal products (g)	190.89	178.11	66.67	307.88	77.09	25.70	**360.09**	**177.23**	**0.56**	**2585.28**	**591.43**	**143.44**	266.67	219.76	93.24	643.45	176.93	58.98
Eggs and egg dishes (g)	14.67	7.00	3.50	39.50	12.16	4.05	**16.35**	**17.50**	**0.00**	**50.00**	**13.60**	**3.30**	16.22	17.50	3.50	50.00	15.30	5.10
Fats and oils (g)	8.43	6.79	1.82	17.42	5.67	1.89	16.08	5.60	0.21	177.12	41.83	10.15	**20.99**	**6.51**	**4.69**	**91.82**	**27.91**	**9.30**
Fish and fish products (g)	18.40	16.24	0.00	46.48	16.11	5.37	**33.92**	**27.23**	**4.20**	**101.00**	**24.78**	**6.01**	23.69	19.32	8.12	47.46	14.26	4.75
Fruit (g)	179.28	191.55	77.25	290.05	78.72	26.24	186.10	135.20	47.95	1047.35	228.21	55.35	**243.75**	**67.90**	**17.50**	**1477.80**	**468.90**	**156.30**
Meat and meat products (g)	136.46	142.58	83.52	183.89	41.33	13.78	**215.23**	**191.01**	**69.05**	**533.97**	**121.41**	**29.45**	108.71	100.55	50.61	156.00	32.42	10.81
Milk and milk products (g)	96.20	90.26	22.82	182.64	46.81	15.60	**126.84**	**89.28**	**4.06**	**423.20**	**118.47**	**28.73**	67.40	57.52	14.98	165.60	54.06	18.02
Non-alcoholic beverages (g)	**758.66**	**656.30**	**196.30**	**1494.14**	**445.84**	**148.61**	588.49	503.00	9.48	1752.20	460.08	111.59	614.78	477.80	166.20	1748.64	488.15	162.72
Nuts and seeds (g)	3.60	2.10	0.00	12.90	3.89	1.30	**23.55**	**2.10**	**0.00**	**137.66**	**39.98**	**9.70**	11.23	2.10	0.00	75.00	24.22	8.07
Potatoes (g)	59.48	59.08	17.50	128.36	33.49	11.16	**125.10**	**83.36**	**0.00**	**392.69**	**113.43**	**27.51**	93.20	83.65	26.39	207.83	53.72	17.91
Soups and sauces (g)	**131.71**	**116.10**	**44.80**	**230.10**	**65.55**	**21.85**	85.88	72.50	0.00	280.40	62.23	15.09	102.51	32.20	23.80	377.90	128.17	42.72
Sugars (g)	34.21	21.44	3.92	133.96	40.57	13.53	68.07	39.22	0.00	530.50	124.68	30.24	**84.49**	**76.37**	**22.90**	**254.77**	**72.12**	**24.04**
preserves and snacks (g)	233.87	224.45	106.01	383.01	105.59	35.20	**270.98**	**218.28**	**20.64**	**747.81**	**182.05**	**44.15**	187.66	174.48	34.86	430.16	143.55	47.85
Vegetables	2.78	5.00	0.00	5.00	2.64	0.88	4.71	5.00	0.00	5.00	1.21	0.29	**5.00**	**5.00**	**5.00**	**5.00**	**0.00**	**0.00**
DII	**1.88**	**2.07**	**−1.24**	**4.71**	**1.86**	**0.62**	1.44	2.01	−3.077	6.52	2.25	0.55	**2.62**	**3.18**	**−2.80**	**6.00**	**2.90**	**0.97**
hs troponin I	**4.79**	**2.70**	**1.30**	**14.10**	**4.43**	**1.48**	1.94	1.80	0.10	7.80	1.86	0.45	1.02	0.80	0.30	2.20	0.59	0.20
TC	5.49	4.60	3.50	10.10	2.19	0.73	**6.01**	**5.90**	**3.90**	**8.40**	**1.24**	**0.30**	5.64	6.00	3.20	7.80	1.55	0.52
HDL-C	1.13	1.10	0.80	1.50	0.26	0.09	1.19	1.20	0.80	1.80	0.29	0.07	**1.21**	**1.30**	**0.90**	**1.50**	**0.20**	**0.07**
LDL-C	3.54	2.60	2.20	7.70	1.82	0.61	**3.84**	**3.80**	**2.20**	**6.00**	**1.01**	**0.24**	3.67	4.10	2.00	5.40	1.29	0.43
TG	1.93	1.80	0.80	3.70	1.01	0.34	**2.31**	**2.40**	**0.80**	**3.60**	**0.64**	**0.16**	1.87	1.50	0.80	4.60	1.16	0.39
TSH	1.76	1.74	0.88	2.77	0.61	0.19	**2.27**	**2.06**	**0.75**	**5.74**	**1.19**	**0.31**	3.31	3.00	2.05	4.79	1.02	0.32
fT4	**13.20**	**13.90**	**9.48**	**15.84**	**2.01**	**0.67**	12.25	12.04	10.19	14.68	1.51	0.37	11.79	12.07	9.32	13.98	1.40	0.47
fT3	3.94	4.21	2.71	4.97	0.69	0.23	4.31	4.27	3.43	5.59	0.55	0.13	**4.68**	**4.64**	**3.75**	**5.57**	**0.65**	**0.22**
Insulin	**18.91**	**16.30**	**4.30**	**31.20**	**9.83**	**3.28**	17.26	15.30	4.50	56.30	11.14	2.70	18.66	18.80	6.80	31.40	8.58	2.86
HOMA-IR	17.90	6.50	1.70	113.40	35.91	11.9710	34.89	33.40	1.40	127.21	32.22	7.81	**43.62**	**35.76**	**11.61**	**87.12**	**26.29**	**8.77**
Weight	**135.48**	**130.70**	**119.40**	**154.70**	**12.23**	**4.08**	125.80	125.60	95.90	183.60	24.20	5.87	123.91	119.80	104.70	145.10	12.66	4.22
SMM	**41.23**	**40.30**	**33.50**	**52.20**	**7.08**	**2.36**	38.17	36.50	26.90	54.20	9.05	2.20	36.13	32.90	30.30	56.20	8.40	2.80
BFM	**62.87**	**64.30**	**43.80**	**78.40**	**11.89**	**3.96**	58.61	55.90	40.40	90.30	13.22	3.21	60.26	62.00	42.20	71.40	10.12	3.37
PBF%	45.30	43.90	34.30	55.90	7.19	2.40	47.64	49.40	33.50	64.50	7.14	1.73	**48.83**	**51.00**	**30.30**	**55.10**	**7.75**	**2.58**
BMI	**47.64**	**49.80**	**36.40**	**57.50**	**6.97**	**2.32**	42.21	41.70	35.40	54.80	5.69	1.38	42.90	42.40	37.70	49.20	3.53	1.18
BMR	**1938.22**	**1903.00**	**1648.00**	**2341.00**	**255.32**	**85.11**	1821.35	1768.00	1418.00	2385.00	321.52	77.99	1745.11	1635.00	1532.00	2462.00	300.39	100.13
W-H ratio	1.05	1.04	0.84	1.17	0.12	0.04	**1.09**	**1.07**	**0.92**	**1.27**	**0.08**	**0.02**	1.06	1.06	0.93	1.22	0.09	0.03
VFL	19.44	20.00	16.00	20.00	1.33	0.44	**19.94**	**20.00**	**19.00**	**20.00**	**0.24**	**0.06**	19.89	20.00	19.00	20.00	0.33	0.11
M1	**680.22**	**788.00**	**197.00**	**890.00**	**238.42**	**79.47**	503.94	423.00	186.00	883.00	257.74	62.51	331.89	255.00	223.00	633.00	156.68	52.23
M2	248.56	177.00	102.00	504.00	143.89	47.96	330.18	357.00	113.00	569.00	152.32	36.94	**412.44**	**410.00**	**249.00**	**575.00**	**111.31**	**37.10**
M3	22.78	10.00	1.00	85.00	28.87	9.62	60.82	54.00	2.00	128.00	40.05	9.71	**91.22**	**87.00**	**40.00**	**140.00**	**32.34**	**10.78**
M4	46.11	2.00	0.00	214.00	82.86	27.62	105.06	80.00	0.00	302.00	97.15	23.56	**164.44**	**184.00**	**24.00**	**285.00**	**92.40**	**30.80**
NDI	1.43	1.21	1.11	2.32	0.43	0.14	1.77	1.76	1.12	2.55	0.47	0.12	**2.09**	**2.22**	**1.54**	**2.46**	**0.34**	**0.11**
FREQ MN TOTAL	**9.94**	**10.00**	**4.00**	**19.00**	**4.72**	**1.57**	8.21	9.00	1.50	14.50	3.87	0.94	4.78	5.00	2.00	7.00	1.73	0.58
FREQ NB TOTAL	3.83	2.00	0.00	18.00	5.42	1.81	**5.56**	**5.00**	**1.00**	**13.00**	**3.80**	**0.920**	5.39	4.00	1.50	14.00	4.10	1.37
FREQ NPB TOTAL	5.61	4.50	1.50	14.50	4.05	1.35	**6.41**	**5.50**	**1.00**	**17.00**	**5.02**	**1.22**	3.78	3.00	2.00	10.00	2.46	0.82
APOPTOSIS	**11.44**	**10.00**	**1.00**	**42.00**	**12.13**	**4.04**	6.82	5.00	2.00	18.00	4.38	1.06	5.00	4.00	3.00	9.00	2.29	0.76
NECROSIS	2.56	0.00	0.00	23.00	7.67	2.56	**2.65**	**0.00**	**0.00**	**29.00**	**7.33**	**1.78**	0.00	0.00	0.00	0.00	0.00	0.00
Net Fpg, TI	5.55	3.03	0.00	15.06	5.55	1.85	5.22	3.14	0.23	13.01	3.97	0.96	**6.63**	**5.30**	**1.41**	**13.50**	**4.37**	**1.46**
TI	**12.02**	**10.53**	**6.65**	**17.72**	**3.84**	**1.28**	10.28	10.04	4.61	17.88	4.12	0.998	8.97	8.52	4.50	15.93	3.58	1.193
Age	**55.69**	**56.00**	**51.00**	**59.00**	**2.76**	**0.92**	45.43	45.80	40.26	51.41	3.58	0.869	32.95	35.31	26.35	38.73	4.95	1.65

**Table 6 nutrients-15-00899-t006:** Values of three DNA damage assays’ parameters for severely obese who previously had tumors (n = 8) and the rest of the obese group (NonT, n = 45); for severely obese who have family history of tumors (FMT, n = 25) and the rest of the obese group (NT, n = 28).

DNA Damage Parameters	T/NT	Mean	Median	Min	Max	SD	SE
MN	NonT	8.76	7.00	1.50	24.00	5.09	0.76
	T	10.38	9.75	3.00	17.50	5.30	1.88
NB	NonT	6.03	4.00	0.00	21.00	5.29	0.79
	T	3.19	2.00	0.50	11.00	3.40	1.20
NPB	NonT	6.02	4.50	0.50	17.00	4.44	0.66
	T	4.13	3.00	1.00	10.50	3.23	1.14
APOPTOSIS	NonT	7.84	7.00	1.00	42.00	6.88	1.02
	T	8.25	7.00	0.00	26.00	8.29	2.93
NECROSIS	NonT	2.44	0.00	0.00	33.00	7.39	1.10
	T	4.75	1.00	0.00	23.00	8.01	2.83
Net Fpg, TI, %	NonT	6.45	4.73	0.00	31.59	5.94	0.89
	T	5.29	2.57	0.00	24.06	7.93	2.80
TI, %	NonT	10.32	9.98	4.50	17.88	3.83	0.57
	T	11.56	12.74	4.68	18.27	5.02	1.77
AGE, years	NonT	49.68	50.00	26.35	68.00	11.39	1.70
	T	59.42	64.00	41.00	67.00	9.47	3.35
MN	NT	8.3	7.0	3.0	17.5	4.2	0.80
	FMT	9.7	9.3	1.5	24.0	5.9	1.16
NB	NT	4.7	3.0	0.5	20.5	4.4	0.85
	FMT	6.5	4.5	0.0	21.0	5.7	1.12
NPB	NT	5.5	3.5	0.5	16.0	4.3	0.83
	FMT	6.0	4.8	1.0	17.0	4.4	0.85
APOPTOSIS	NT	7.4	6.0	2.0	26.0	5.4	1.05
	FMT	8.5	7.0	0.0	42.0	8.4	1.65
NECROSIS	NT	2.7	0.0	0.0	33.0	7.8	1.49
	FMT	2.9	0.0	0.0	29.0	7.3	1.43
Net Fpg, TI, %	NT	6.0	3.0	0.0	31.6	6.6	1.28
	FMT	6.5	5.0	0.0	24.1	5.8	1.15
TI, %	NT	10.0	9.6	4.5	17.9	3.6	0.70
	FMT	11.0	11.6	4.6	18.3	4.4	0.86
AGE	NT	48.9	49.0	26.4	68.0	12.5	2.41
	FMT	53.5	56.5	28.9	67.0	10.3	2.01

MN—micronuclei frequency per 1000 binucleated cells (BN), NB—nuclear buds frequency per 1000 BN, NPB—nucleoplasmic frequency per 1000 BN, TI—tail intensity.

**Table 7 nutrients-15-00899-t007:** Demonstration of the individual levels of DNA damage for both comet and CBMN assay for individuals with previous tumors and the ones with family history of tumors.

Individuals with previous tumors					
**Individual No.**	**>9% TI**	**>12.4% TI**	**>13 MN**	**>10 NPB**	**>5 NB**
1	+	+	+		
2	+		+		
3	+	+			
4			+	+	
5	+	+			
6	+	+			
7	+	+			
8			+		
Individuals with family history of tumors					
**Individual No.**	**>9% TI**	**>12.4% TI**	**>13 MN**	**>10 NPB**	**>5 NB**
1					
2	+	+	+		
3	+	+			
4	+	+	+		
5	+	+		+	
6			+	+	
7	+	+			
8			+	+	
9	+	+			
10	+	+		+	+
11	+	+			
12	+	+			+
13	+	+			+
14	+	+		+	+
15	+	+			
16	+	+		+	+
17					+
18	+	+	+		+
19	+	+			
20					
21			+		+
22					+
23			+		+
24					+
25					+

## Data Availability

Not applicable. All data are in the manuscript.

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
