# Peer review of "Combined Approach: FFQ, DII, Anthropometric, Biochemical and DNA Damage Parameters in Obese with BMI ≥ 35 kg m−2"

_nutrients, 2023, doi:10.3390/nu15040899_

Round 1

Reviewer 1 Report

Milic and coll aimed to analyze the potential relationship between DNA damage, biochemical and anthropometrical parameters in obese patients.
The influence of diet and DII was also observed. This study appears rich of data and correlations, but  most of them  are redondant and not innovative. According this revisor the manuscript need of major revision in order to eliminate the many criticisms.

-Firstly, the text is too long. The introduction must be strongly reduced. It appears like a review. Indeed the authors should highligh only the most recent references supporting their work hypothesis and to delete all that is well known in the pathophysiology of obesity. 

- the description of the results is highly confusing. Most results reported in the text can be easily extrapolated by the numerous tables. Then,   it is suggested to reduce the results, by abolishing all numerical data and leaving only the comments on the highlighted correlations.

- this revisor also noted a critical point regarding the study design. It's known that oxidative stress and inflammation are pathological conditions underling the metabolic dysfunctions occurring in obesity. In the same time, these pathological conditions are involved in the onset and progression of tumor transformations. Being both involved in the pathogenesis of obesity as well as in cancer development it is not clear the innovation of the results. The authors should comment in a clearer manner on the common pathogenesis of both diseases by highlighting the novelty and the scientific progression obteined by the study

- In the methods, it is described that the recruited volunteers were 53 and 25 of them had closer family relatives with tumor/cancer disease and 8 had a tumor surgically removed in the past. Have this revisor well understand ? Becouse if it is so , all data are to be revised taking into account the falily story of tumors for the 50% of the partecipants.

- what were the inclusion citeria for the healthy non-obese patients?

Author Response

The Authors would like to thank the Reviewer #1 for her/his valuable comments regarding our manuscript. We have responded to all of the comments and revised the paper accordingly. Detailed responses to each comment are provided in the attachment

Reviewer 2 Report

Milic et al. studied the genome stability in 53 severely obese subjects by using micronucleus test, alkaline, and Fpg comet assays. In addition, they compared the results of genomic stability measurements with biochemical parameters, nutrient intake, and anthropometric parameters. They have a small group of subjects, but they have gained detailed information on the blood biochemical parameters and nutritional intake of these subjects. The results are presented well, and the manuscript is well-written. Although there are several publications in this direction, they revisited the effect of obesity on genomic stability and contributed by measuring detailed biochemical parameters and collecting detailed information regarding lifestyle factors and nutritional preferences. However, huge tables with several factors are reducing the readability of this manuscript and the correlation analysis with several factors sometimes without discussing the mechanism or the link between the factors is making hard to understand the main message of this manuscript. Since the correlation between variables does not automatically mean that the observed change in one variable is the cause of the observed change/effect in the second variable! That is why I believe that authors should be careful during the discussion of correlation data, especially in case of a weak correlation as in Fpg comet assay and dietary inflammatory index. 

Suggestions:

P3, line 118, typo › measurements

P7, alkaline, and Fpg comet assay method description, it is not clear whether the authors used whole blood samples or isolated peripheral blood mononuclear cells. More methodological details would be helpful for the readers!

P7: did authors use fresh samples or cryopreserved samples in the comet assay and micronucleus test? Did they stimulate the blood cells for the selected endpoints? If they conducted these endpoints in several runs, did they run assay controls? If yes, I believe adding the assay controls might be useful.

The reference values for biochemical blood parameters are clear, however, the reference values for the micronucleus test and comet assay are not clear until the discussion. In the discussion, the authors mentioned that they previously established the reference range for MN frequency. Could authors mention these in the methods/results too? In addition, some more details regarding these reference groups might be useful. For example, age range, BMI etc.

Spearman correlation coefficient is mostly <0.5. Although there is a significant correlation between net Fpg-sensitive sites and inflammatory food index, with a correlation coefficient < 0.5, there is a poor correlation. I suggest the authors mention the strength of the association in results and discussion.

As the authors indicated in the discussion, there is no clear evidence that certain food groups have an impact on obesity. Although a detailed analysis of food groups is interesting, it is hard to understand what the difference in caloric intake is. If there is a significant difference among individuals regarding the daily caloric intake, it might be worth to analyze the correlation between the caloric intake and some parameters like insulin/glucose tolerance, and genomic stability markers.

I understand the fact that long titles are useful to describe the study, but a shorter version might be more attractive for the readers!

As last the authors mentioned that they had eight subjects with a tumor history. I wonder whether these eight individuals who are with higher genomic instability compared to the rest of the group. I believe it is hard to exclude the effect of tumor history on genomic stability, reflecting results from these eight subjects extra would be informative.   

Author Response

The Authors would like to thank the Reviewer #2 for her/his valuable comments regarding our manuscript. We have responded to all of the comments and revised the paper accordingly. Detailed responses to each comment are provided in the attachment.

Round 2

Reviewer 1 Report

The authors have strongly revised the manuscript according the revisor's comment. The manuscript appears more clear and easily to read.